# BiDoRA: Bi-level Optimization-Based Weight-Decomposed Low-Rank Adaptation

**Peijia Qin**                                                                 *pqin@ucsd.edu*
*University of California, San Diego*

**Ruiyi Zhang**                                                               *ruz048@ucsd.edu*
*University of California, San Diego*

**Pengtao Xie**                                                               *p1xie@ucsd.edu*
*University of California, San Diego*

**Reviewed on OpenReview:** *https://openreview.net/forum?id=v2xCm3VYl4*

## Abstract

Parameter-efficient fine-tuning (PEFT) is a flexible and efficient method for adapting large language models (LLMs) to downstream tasks. Among these methods, weight-decomposed low-rank adaptation (DoRA) is a promising approach that decomposes weight matrices into magnitude and direction components to mimic full fine-tuning (FT) better. However, DoRA's simultaneous optimization of these components makes it over-expressive, increases the *risk of overfitting*, and creates a *coupled updating pattern* that limits its learning capacity. To address these issues, we propose **Bi**-level Optimization-Based Weight-**D**ecomposed **Lo**w-**R**ank **A**daptation (**BiDoRA**), a novel PEFT method based on a *bi-level optimization framework*. BiDoRA fundamentally differs from DoRA by optimizing the magnitude and direction in two separate, asynchronous loops using distinct training and validation data splits. This decoupled optimization process effectively mitigates overfitting and allows for more flexible updates that align even more closely with FT. For instance, weight decomposition analysis shows BiDoRA achieves a magnitude-direction update correlation of $-\mathbf{8.042}$, significantly closer to the FT ideal compared to $-\mathbf{1.784}$ for DoRA. Evaluation of BiDoRA on diverse tasks spanning natural language understanding, generation, token classification, and extremely small biomedical datasets reveals that it consistently outperforms DoRA and a wide range of leading PEFT methods. This improvement is statistically significant, as demonstrated on the GLUE benchmark where BiDoRA surpasses DoRA with a p-value of $\mathbf{2.4 \times 10^{-4}}$ in terms of the Wilcoxon signed-rank test. The code for BiDoRA is available at https://github.com/t2ance/BiDoRA.

## 1 Introduction

Large language models (LLMs) (Radford et al., 2019; Brown et al., 2020) have achieved state-of-the-art results across a broad range of NLP tasks, from natural language understanding (NLU) (Wang et al., 2019) to natural language generation (NLG) (Novikova et al., 2017). Parameter-efficient fine-tuning (PEFT) methods (Houlsby et al., 2019; Hu et al., 2022b) have been introduced as a promising solution for adapting LLMs for downstream data. PEFT approaches update only a subset of the pre-trained parameters, achieving performance comparable to full-finetuning (FT) while requiring significantly fewer computational resources.

One popular type of PEFT is low-rank adaptation (LoRA, Hu et al. (2022b)), which attaches low-rank matrices to the pre-trained weights and updates only these matrices during fine-tuning. Liu et al. (2024a) shows that when decomposing the weights into magnitude and direction, their correlation (See Appendix D) tends to be positive in LoRA, whereas it is negative in FT. To bridge the training pattern distinction, they

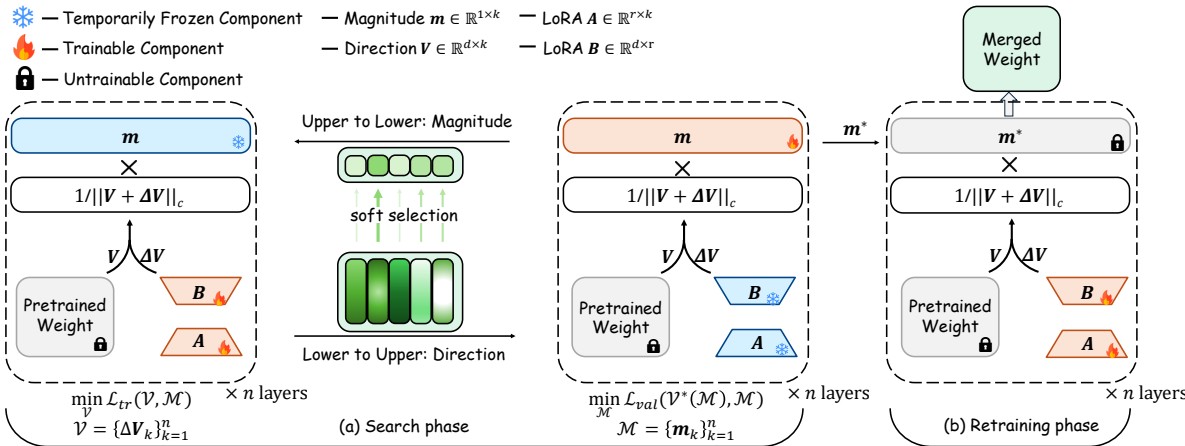

Figure 1: **An overview of BiDoRA.** BiDoRA performs PEFT using a BLO framework. **At the lower level**, BiDoRA learns the direction component $\Delta\mathbf{V}$ of the update matrices using the training split of the downstream dataset. **At the upper level**, BiDoRA optimizes the magnitude component $\mathbf{m}$ with optimized $\Delta\mathbf{V}$ from the lower level, using the validation split of the dataset. After determining the optimal magnitude, the direction component undergoes further fine-tuning on a combined set of both training and validation splits to maximize overall performance.

introduce an explicit reparameterization of the pre-trained weights matrix. The method, named DoRA, decomposes the weights into the column-wise product of magnitude and direction, which determines the direction and magnitude of the weight update, respectively. This approach enables DoRA to share similar learning patterns with FT, thereby outperforming LoRA in multiple tasks. Nonetheless, DoRA introduces **additional parameters** and **over-expressive architecture** compared to LoRA, which can exacerbate overfitting issues when adapting to small downstream datasets (See Table 3). Furthermore, in DoRA, the magnitude and direction components are optimized concurrently, leading to a **constrained updating pattern** due to shared optimization setup (e.g., learning rate, optimizer, batch size).

To address the challenges above, we propose BiDoRA, a **Bi**-level Optimization-Based Weight-**D**ecomposed **Lo**w-**R**ank **A**daptation method for PEFT. BiDoRA facilitates an even more flexible updating pattern and mitigates overfitting by separately optimizing the two components on different data splits with distinct optimization levels. BiDoRA is based on a bi-level optimization (BLO) framework: At the lower level, the low-rank direction component is updated using the training split, while the magnitude component remains fixed. At the upper level, the magnitude component is updated by minimizing the loss on the validation split via hypergradient descent. Subsequently, the direction component is further fine-tuned with the optimal magnitude frozen to maximize the performance. These two optimization steps are performed iteratively until convergence. Fig. 1 provides an overview of BiDoRA.

A similar strategy of combating overfitting based on BLO has been utilized in the well-established practice of differentiable neural architecture search (DARTS, Liu et al. (2019)), where architecture and sub-networks are learned using different dataset splits. Optimizing the selection variables and sub-networks in a single loop can result in an over-expressive network since the selection variables tend to select all sub-networks to achieve the best expressiveness, which, however, incurs severe overfitting. In contrast, training the sub-networks with the selection module fixed on the training split while validating the effectiveness of the selection module on the unseen validation split effectively eliminates the risk of overfitting. Similarly, we treat the **magnitude component as the architecture** and the **direction component as the sub-networks** and train these components on **separate datasets**. As shown in Table 3, BiDoRA demonstrates better resistance to overfitting compared to DoRA, given the smaller performance gap between the training set and test set. Furthermore, the asynchronous gradient update steps at the two optimization levels in BiDoRA facilitate better decoupling of the two components, leading to a more flexible update pattern that closely resembles FT. As illustrated in Fig. 3, the updates across different layers using BiDoRA have a correlation

value that is closest to that of FT, highlighting its superior learning capability compared to both DoRA and LoRA.

Our work makes the following key contributions:

- We propose **BiDoRA**, a novel PEFT method based on bi-level optimization. In contrast to DoRA, which trains the magnitude and direction components on a single dataset, BiDoRA optimizes these components at different optimization levels.

- Our strategy effectively mitigates the risk of overfitting and results in a parameter update pattern that more closely resembles full fine-tuning.

- Extensive experiments on various downstream tasks highlight the superior performance of BiDoRA. BiDoRA consistently surpasses several baseline methods, including LoRA and DoRA.

## 2 Related Work

### 2.1 Parameter Efficient Fine-Tuning Methods

Parameter-efficient fine-tuning (PEFT) methods aim to reduce the high costs associated with full fine-tuning large-scale models by updating only a relatively small subset of pre-trained parameters, rather than the entire model, to adapt to downstream tasks. Existing PEFT methods can be mainly categorized into three types.

**The first category, known as adapter-based methods,** injects additional trainable modules into the original frozen backbone. For instance, Houlsby et al. (2019) suggests adding linear modules in sequence to existing layers, while He et al. (2022) proposes integrating these modules in parallel with the original layers to enhance performance. Recent advances include SAN (Xu et al., 2023), FADA (Bi et al., 2024), and SET (Yi et al., 2024). SAN presents a side adapter network attached to a frozen CLIP model, which contains two branches for predicting mask proposals and attention biases. FADA introduces a frequency-adapted learning scheme that uses the Haar wavelet transform to decompose frozen features into low- and high-frequency components, which are processed separately to enhance domain generalization. SET proposes a spectral-decomposed token learning framework that leverages the Fast Fourier Transform to separate frozen features into amplitude and phase components, enhancing them with spectral tokens and attention optimization.

**The second category is prompt tuning methods**, which add extra soft tokens (prompts) to the initial input. During the fine-tuning stage, only these trainable soft tokens are updated, as demonstrated in works such as Lester et al. (2021) and Razdaibiedina et al. (2023). Unfortunately, the first two categories lead to increased inference latency compared to fully fine-tuned models.

**The third prominent category focuses on low-rank adaptation**, pioneered by LoRA (Hu et al., 2022a). LoRA injects trainable, low-rank matrices into a model's layers, freezing the original weights. A key advantage is that these low-rank updates can be merged into the original weights before inference, thus incurring no additional latency. Subsequent works have aimed to improve LoRA's efficiency and performance. For instance, AdaLoRA (Zhang et al., 2023) dynamically reallocates the parameter budget based on the importance scores of weight matrices. Zhang et al. (2024b) uses meta-learning to search for the optimal rank of LoRA matrices, further improving its performance on downstream tasks. Pushing parameter efficiency further, VeRA (Kopiczko et al., 2024) employs a single pair of shared low-rank matrices across all layers, while AFLoRA (Liu et al., 2024b) freezes a portion of adaptation parameters based on a learned score. A distinct sub-direction has emerged that performs adaptation in the frequency domain, including FourierFT (Gao et al., 2024), LaMDA (Azizi et al., 2024), SSH (Shen et al., 2025b), and MaCP (Shen et al., 2025a). These methods learn updates in transformed spectral spaces, such as the Fourier, discrete Hartley, or discrete cosine domains, rather than directly in the weight space. Other research has focused on bridging the performance gap between LoRA and full fine-tuning. Liu et al. (2024a) found that LoRA's update patterns differ significantly from full fine-tuning, potentially constraining its learning capacity. To mitigate this, they proposed DoRA (Liu et al., 2024a), which decomposes pre-trained weights into magnitude and direction components and uses LoRA for efficient directional updates, better mimicking full fine-tuning.

## 2.2 Bi-level Optimization

Bi-level optimization (BLO) has been widely applied in various machine learning tasks, including meta-learning (Finn et al., 2017; Rajeswaran et al., 2019), neural architecture search (NAS) (Liu et al., 2019; Zhang et al., 2021), and hyperparameter optimization (Lorraine et al., 2020; Franceschi et al., 2017). Despite its wide usage, solving BLO problems can be challenging due to the inherent nature of nested optimization problems. Several algorithms have been proposed to address this challenge, including zeroth-order methods such as Bayesian optimization (Cui & Bai, 2019) and first-order algorithms based on hypergradients (Pearlmutter & Siskind, 2008; Lorraine et al., 2020). Among these approaches, gradient-based BLO has received significant attention because it can scale to high-dimensional problems with a large number of trainable parameters.

Inspired by NAS, where a bi-level approach is used to learn an architecture and its sub-network weights on separate data splits to prevent overfitting, we adapt the BLO framework to parameter-efficient fine-tuning (PEFT), specifically for the weight-decomposed adaptation introduced by DoRA. Unlike in NAS, where BLO searches for a network architecture, BiDoRA repurposes it to decouple the optimization of a weight matrix's two components: magnitude and direction. This approach marks a significant departure from previous PEFT methods like LoRA and DoRA, which optimize all trainable parameters simultaneously on a single dataset. In this work, we extend the application of gradient-based BLO to develop a robust and effective PEFT method for pre-trained models. By assigning the magnitude and direction components to different optimization levels with distinct data splits, BiDoRA creates a decoupled, flexible updating pattern that better mitigates overfitting and more closely resembles the learning behavior of full fine-tuning.

## 3 Preliminary

LoRA (Hu et al., 2022b) involves attaching the product of two low-rank matrices to the pre-trained weights and fine-tuning these low-rank matrices on downstream datasets with the pre-trained weights frozen. It is based on the assumption that parameter updates made during fine-tuning exhibit a low intrinsic rank. Formally, given a pre-trained weight matrix $\mathbf{W_0} \in \mathbb{R}^{d \times k}$, LoRA attaches a low-rank update matrix $\Delta \mathbf{W} \in \mathbb{R}^{d \times k}$ to the pre-trained weight. This update matrix can be decomposed as $\Delta \mathbf{W} = \mathbf{BA}$, where $\mathbf{B} \in \mathbb{R}^{d \times r}$ and $\mathbf{A} \in \mathbb{R}^{r \times k}$ are two low-rank matrices, with $r \ll \min(d, k)$. Consequently, the weight matrix $\mathbf{W}'$ is represented as follows:

$$\mathbf{W}' = \mathbf{W_0} + \Delta \mathbf{W} = \mathbf{W_0} + \mathbf{BA} \tag{1}$$

In this setup, only the LoRA matrix $\Delta \mathbf{W}$ is updated. Liu et al. (2024a) found that LoRA and full fine-tuning exhibit different learning patterns by performing weight decomposition on fine-tuned weight matrices (See Appendix D). To bridge this discrepancy, weight-decomposed low-rank adaptation (DoRA, Liu et al. (2024a)) further reparameterizes the weight matrices by explicitly decomposing them into learnable magnitude and direction components. Formally, DoRA performs adaptation as follows:

$$\mathbf{W}' = \mathbf{m} \frac{\mathbf{V} + \Delta \mathbf{V}}{\|\mathbf{V} + \Delta \mathbf{V}\|_c} = \mathbf{m} \frac{\mathbf{W_0} + \mathbf{BA}}{\|\mathbf{W_0} + \mathbf{BA}\|_c} \tag{2}$$

where $\Delta \mathbf{V}$ is a product of two learnable low-rank matrices, $\mathbf{B}$ and $\mathbf{A}$, while the magnitude component $\mathbf{m} \in \mathbb{R}^{1 \times k}$ is a learnable vector. Here, $\|\cdot\|_c$ represents the vector-wise norm of a matrix computed across each column, using the $L_2$ norm. In DoRA, both components are optimized concurrently on a single downstream dataset. In this work, we aim to improve DoRA by further decoupling the training of the two components.

## 4 Methods

### 4.1 Overview of BiDoRA

Our method, BiDoRA, optimizes the trainable parameters in DoRA layers by solving a BLO problem. Let $\mathcal{M} = \{\mathbf{m}_1, \mathbf{m}_2, \ldots, \mathbf{m}_n\}$ denote the set of magnitude components for all $n$ DoRA modules, and $\mathcal{V} = \{\Delta \mathbf{V}_1, \Delta \mathbf{V}_2, \ldots, \Delta \mathbf{V}_n\}$ denote the set of corresponding direction components. Specifically, we first learn the direction components $\mathcal{V}^*(\mathcal{M})$ on the training split of the downstream dataset $\mathcal{D}_{tr}$ at the lower level. The

magnitude component $\mathcal{M}$ is tentatively fixed at this level; thus, the resulting optimal direction component $\mathcal{V}^*(\mathcal{M})$ is a function of $\mathcal{M}$. At the upper level, we determine the optimal magnitude component $\mathcal{M}^*$ by optimizing the loss on a validation split $\mathcal{D}_{val}$. In practice, $\mathcal{D}_{tr}$ and $\mathcal{D}_{val}$ are typically created by splitting the original training set without using additional data. This BLO problem is solved using an efficient gradient-based algorithm, where parameters at two levels are optimized iteratively until convergence. While this work focuses on the empirical validation of BiDoRA, our choice of optimization strategy is grounded in established theoretical research. The convergence properties of similar gradient-based bi-level algorithms have been previously analyzed (Pedregosa, 2016; Rajeswaran et al., 2019), providing confidence in the stability of our training procedure. Furthermore, the ability of such frameworks to improve generalization—a core objective of BiDoRA—has also been formally studied (Bao et al., 2021), supporting the rationale that our approach can mitigate overfitting.

## 4.2 Orthogonal Regularization

A central goal of BiDoRA is to learn the two disentangled components of a weight update: magnitude and direction. The direction component, $\Delta\mathbf{V}$, is responsible for finding a low-rank basis for the update directions. To maximize the expressive power of this component and prevent overfitting, its basis vectors (i.e., the columns of the direction matrix) should be as diverse and non-redundant as possible.

The orthogonality of neural network weights has been identified as a beneficial property (Bansal et al., 2018) and can effectively mitigate the overfitting issue (Balestriero & richard baraniuk, 2018). By enforcing orthogonality, the direction vectors are constrained to represent distinct, independent pathways for updates. This ensures that the limited parameter budget of the low-rank matrix is used efficiently to explore the solution space. Therefore, we define a Gram regularization loss (Xie et al., 2017) for the direction component:

$$\mathcal{R}(\mathcal{V}) = \sum_{k=1}^{n} \left\| (\mathbf{V}_k + \Delta\mathbf{V}_k)^\top (\mathbf{V}_k + \Delta\mathbf{V}_k) - \mathbf{I} \right\|_F^2 \tag{3}$$

where $\mathbf{I}$ is the identity matrix and $\|\cdot\|_F$ denotes the Frobenius norm. Intuitively, $\mathcal{R}(\mathcal{V})$ encourages each column of the direction matrix, representing a specific direction, to be orthogonal to one another. Since each column has already been normalized (equivalent to projected to the unit sphere), this also prompts each column to be far away from the other, thereby reducing the redundancy of parameters. The effectiveness of this constraint is empirically validated in our ablation study (See Table 5), which shows a consistent performance improvement resulting from the enhanced generalization ability of the learned direction component.

## 4.3 A Bi-level Optimization Framework

**Lower level.** At the lower level, we train the low-rank direction component $\mathcal{V}$ by minimizing a loss $\mathcal{L}_{tr}$ defined on the training set $\mathcal{D}_{tr}$. The overall training objective at this level is $\mathcal{L}_{tr}(\mathcal{V}, \mathcal{M}) = \mathcal{L}(\mathcal{V}, \mathcal{M}; \mathcal{D}_{tr}) + \gamma\mathcal{R}(\mathcal{V})$. Here, $\mathcal{L}$ represents the fine-tuning loss, given the low-rank direction component $\mathcal{V}$, the magnitude component $\mathcal{M}$, and the training split $\mathcal{D}_{tr}$ of the downstream dataset. $\mathcal{R}(\mathcal{V})$ is the orthogonal regularizer defined in Eq. (3), with $\gamma$ as a trade-off hyperparameter. In this level, we only update $\mathcal{V}$ while keeping $\mathcal{M}$ fixed, resulting in the following optimization problem:

$$\mathcal{V}^*(\mathcal{M}) = \arg\min_{\mathcal{V}} \mathcal{L}_{tr}(\mathcal{V}, \mathcal{M}) \tag{4}$$

where $\mathcal{V}^*(\mathcal{M})$ denotes the optimal solution for $\mathcal{V}$ in this problem, which is a function of $\mathcal{M}$.

**Upper level.** At the upper level, we validate the previously fixed magnitudes $\mathcal{M}$ on the validation set $\mathcal{D}_{val}$, using the optimal direction component $\mathcal{V}^*(\mathcal{M})$ that was learned at the lower level. This results in a validation loss $\mathcal{L}_{val}(\mathcal{V}^*(\mathcal{M}), \mathcal{M}) = \mathcal{L}(\mathcal{V}^*(\mathcal{M}), \mathcal{M}; \mathcal{D}_{val})$. We determine the optimal magnitude component $\mathcal{M}$ by minimizing this validation loss:

$$\min_{\mathcal{M}} \mathcal{L}_{val}(\mathcal{V}^*(\mathcal{M}), \mathcal{M}) \tag{5}$$

---

**Algorithm 1:** BiDoRA

---

**Input:** Training dataset $\mathcal{D}_{tr}$ and validation dataset $\mathcal{D}_{val}$

**1** Initialize trainable magnitude components $\mathcal{M} = \{\mathbf{m}_k\}_{k=1}^n$ and low-rank direction components
   $\mathcal{V} = \{\Delta\mathbf{V}_k\}_{k=1}^n = \{\{\mathbf{A}_k\}_{k=1}^n, \{\mathbf{B}_k\}_{k=1}^n\}$

**2** // Search Phase

**3 while** *not converged* **do**

**4**  |  Update magnitude $\mathcal{M}$ by descending $\nabla_{\mathcal{M}}\mathcal{L}_{val}(\mathcal{V} - \xi\nabla_{\mathcal{V}}\mathcal{L}_{tr}(\mathcal{V}, \mathcal{M}), \mathcal{M})$

**5**  |  Update direction $\mathcal{V}$ by descending $\nabla_{\mathcal{V}}\mathcal{L}_{tr}(\mathcal{V}, \mathcal{M})$

**6** Derive the optimal magnitude $\mathcal{M}^* = \{m_k^*\}_{k=1}^n$

**7** // Retraining Phase

**8** Train $\mathcal{V}$ until converge using $\mathcal{D}_{tr} \bigcup \mathcal{D}_{val}$ and derive the optimal direction $\mathcal{V}^*$

**Output:** $\mathcal{V}^*$ and $\mathcal{M}^*$

---

**A bi-level optimization framework.** Integrating the two levels of optimization problems, we have the following BLO framework:

$$\min_{\mathcal{M}} \; \mathcal{L}_{val}(\mathcal{V}^*(\mathcal{M}), \mathcal{M})$$
$$s.t. \quad \mathcal{V}^*(\mathcal{M}) = \arg\min_{\mathcal{V}} \; \mathcal{L}_{tr}(\mathcal{V}, \mathcal{M}) \tag{6}$$

Note that these two levels of optimization problems are mutually dependent on each other. The solution of the optimization problem at the lower level, $\mathcal{V}^*(\mathcal{M})$, serves as a parameter for the upper-level problem, while the optimization variable $\mathcal{M}$ at the upper level acts as a parameter for the lower-level problem. By solving these two interconnected problems jointly, we can learn the optimal magnitude component $\mathcal{M}^*$ and incremental direction matrices $\mathcal{V}^*$ in an end-to-end manner.

Two reasons exist behind the choice of setting the magnitude component as the upper level instead of the converse one: 1) In literature, the upper level usually has fewer parameters than the lower level. In our case, the design of setting the magnitude of complexity $\mathcal{O}(k)$ as the upper level and the direction of complexity $\mathcal{O}(dr + kr)$ as the lower level is consistent with the common practice. 2) BiDoRA resembles the DARTS method (Liu et al., 2019) in NAS literature, where the subnets are selected by a selection variable. Specifically, the magnitude vector resembles a selection variable on the direction matrix by softly selecting each direction (subnets) via scaling.

**Optimization algorithm.** We use a gradient-based optimization algorithm (Choe et al., 2023b) to solve the BLO problem presented in Eq. (6). A significant challenge in this process is the computation of the upper-level loss gradient with respect to the magnitude component $\mathcal{M}$, as this gradient depends on the optimal solution of the lower-level problem, $\mathcal{V}^*(\mathcal{M})$. For deep neural networks, the lower-level objective is non-convex, meaning that finding the true optimal solution $\mathcal{V}^*(\mathcal{M})$ would require running its optimization process to full convergence. Performing this complete inner optimization for every single update of the upper-level variable $\mathcal{M}$ is computationally intractable.

To address this issue, we use the following one-step-unrolled approximation of $\mathcal{V}^*(\mathcal{M})$ inspired by previous work (Liu et al., 2019):

$$\nabla_{\mathcal{M}}\mathcal{L}_{val}(\mathcal{V}^*(\mathcal{M}), \mathcal{M}) \approx \nabla_{\mathcal{M}}\mathcal{L}_{val}(\mathcal{V} - \xi\nabla_{\mathcal{V}}\mathcal{L}_{tr}(\mathcal{V}, \mathcal{M}), \mathcal{M}) \tag{7}$$

where $\xi$ is the learning rate at the lower level, and the one-step-unrolled model $\bar{\mathcal{V}} = \mathcal{V} - \xi\nabla_{\mathcal{V}}\mathcal{L}_{tr}(\mathcal{V}, \mathcal{M})$ is used as a surrogate for the optimal solution $\mathcal{V}^*(\mathcal{M})$. We then compute the approximated gradient as

follows:

$$\nabla_{\mathcal{M}}\mathcal{L}_{val}(\mathcal{V} - \xi\nabla_{\mathcal{V}}\mathcal{L}_{tr}(\mathcal{V}, \mathcal{M}), \mathcal{M})$$

$$= \nabla_{\mathcal{M}}\mathcal{L}_{val}(\bar{\mathcal{V}}, \mathcal{M}) - \xi\nabla^2_{\mathcal{M}, \mathcal{V}}\mathcal{L}_{tr}(\mathcal{V}, \mathcal{M})\nabla_{\bar{\mathcal{V}}}\mathcal{L}_{val}(\bar{\mathcal{V}}, \mathcal{M}) \tag{8}$$

$$\approx \nabla_{\mathcal{M}}\mathcal{L}_{val}(\bar{\mathcal{V}}, \mathcal{M}) - \xi\frac{\nabla_{\mathcal{M}}\mathcal{L}_{tr}(\mathcal{V}^+, \mathcal{M}) - \nabla_{\mathcal{M}}\mathcal{L}_{tr}(\mathcal{V}^-, \mathcal{M})}{2\epsilon} \tag{9}$$

where $\epsilon$ is a small scalar and $\mathcal{V}^{\pm} = \mathcal{V} \pm \epsilon\nabla_{\bar{\mathcal{V}}}\mathcal{L}_{val}(\bar{\mathcal{V}}, \mathcal{M})$. Since directly computing the matrix-vector multiplication term in Eq. (8) is computationally expensive, we use finite difference to approximate this product as in Eq. (9), following Liu et al. (2019). As detailed in Algorithm 1, the direction component $\mathcal{V}$ and the magnitude component $\mathcal{M}$ are updated using gradient descent iteratively until convergence. After acquiring the optimal magnitudes $\mathcal{M}^*$ through the process above, the direction component $\mathcal{V}$ is retrained on the union of training and validation splits to achieve the best performance on downstream tasks, resulting in the final learned $\mathcal{V}^*$. All splits are intentionally used during retraining to maximize data utilization and performance.

In practice, the convergence of the search phase is determined by the evaluation metric at the upper level. For the subsequent retraining phase, we adopt a stopping criterion similar to DoRA's, observing performance on a separate, held-out test set that is not used during training.

Table 1: RoBERTa$_{\text{base/large}}$ (R$_{\text{b/l}}$) and DeBERTa$_{\text{XXL}}$ (D$_{\text{XXL}}$) with different fine-tuning methods on the GLUE benchmark (Wang et al., 2019). A higher value is better for all datasets. The best results are shown in **bold**.

| Method | #Params | MNLI | SST-2 | MRPC | CoLA | QNLI | QQP | RTE | STS-B | Avg. |
|---|---|---|---|---|---|---|---|---|---|---|
| R$_{\text{b}}$(FT) | 125.0M | 90.3 | 94.8 | 89.3 | 61.6 | 86.7 | 92.8 | 76.9 | 91.2 | 85.5 |
| R$_{\text{b}}$(Adapter) | 0.9 M | 86.5 | 94.0 | 88.4 | 58.8 | 92.5 | 89.1 | 71.2 | 89.9 | 83.8 |
| R$_{\text{b}}$(LoRA) | 0.15 M | 86.8 | 94.3 | 88.0 | 60.3 | **93.0** | 89.6 | 72.9 | 90.1 | 84.4 |
| R$_{\text{b}}$(DoRA) | 0.17 M | 86.8 | 94.2 | 89.2 | 60.5 | 92.9 | 89.6 | 73.2 | **90.2** | 84.6 |
| R$_{\text{b}}$(BiDoRA) | 0.17 M | **87.1** | **94.4** | **89.4** | **61.3** | 92.7 | **90.6** | **76.0** | 90.1 | **85.2** |
| R$_{\text{l}}$(FT) | 355.0M | 90.2 | 96.4 | 90.9 | 68.0 | 94.7 | 92.2 | 86.6 | 92.4 | 88.9 |
| R$_{\text{l}}$(Adapter) | 0.8M | 90.3 | 96.3 | 87.7 | 66.3 | **94.7** | 91.5 | 72.9 | 91.5 | 86.4 |
| R$_{\text{l}}$(LoRA) | 0.39 M | **90.6** | 96.3 | 90.0 | 66.9 | 94.5 | 91.2 | 86.3 | 91.7 | 88.4 |
| R$_{\text{l}}$(DoRA) | 0.39 M | **90.6** | 96.4 | 89.8 | 65.8 | **94.7** | 91.2 | 86.6 | **92.0** | 88.4 |
| R$_{\text{l}}$(BiDoRA) | 0.39 M | **90.6** | 96.1 | **90.1** | 67.0 | 94.6 | **91.7** | 86.9 | 92.0 | **88.6** |
| D$_{\text{XXL}}$(DoRA) | 4.9M | 91.2 | **96.3** | 92.3 | 71.1 | **95.3** | 91.6 | 91.8 | **90.8** | 90.0 |
| D$_{\text{XXL}}$(BiDoRA) | 4.9M | **91.7** | 96.3 | **92.6** | **72.3** | 95.2 | **92.0** | **92.3** | 90.8 | **90.4** |

## 5 Experiments

### 5.1 Experimental Setup

We compare BiDoRA with several PEFT methods, including Full Fine-Tuning (FT), Adapter Tuning (Houlsby et al., 2019; Lin et al., 2020; Rücklé et al., 2021; Pfeiffer et al., 2021), LoRA (Hu et al., 2022a), A AdaLoRA (Zhang et al., 2023), DoRA (Liu et al., 2024a), VeRA (Kopiczko et al., 2024), FourierFT (Gao et al., 2024), AFLoRA (Liu et al., 2024b), LaMDA (Azizi et al., 2024), SSH (Shen et al., 2025b), MaCP (Shen et al., 2025a). BiDoRA **does not use any additional data** compared to other baselines, as we create the validation set for upper-level optimization by splitting the original training set with an 8:2 ratio for all tasks. Detailed descriptions of these baseline methods are provided in Appendix C.

Our experiments cover a wide range of tasks, including natural language understanding (Section 5.2), extremely small biomedical datasets (Section 5.3), natural language generation (Appendix H.1), and token

classification (Appendix H.2). Please refer to the detailed dataset settings and experimental settings in Appendix A and Appendix B, respectively. Our implementation is based on the Huggingface Transformers library (Wolf et al., 2019) and the Betty library (Choe et al., 2023b).

### 5.2 Experiments on Natural Language Understanding Tasks

In this section, we evaluate the performance of BiDoRA on NLU tasks.

**Main results.** Table 1 presents the results of fine-tuning the RoBERTa-base, RoBERTa-large, and De-BERTa XXL models on the GLUE benchmark with baseline PEFT methods and BiDoRA. The results show that BiDoRA achieves superior or comparable performance compared to baseline methods across all datasets with the same number of trainable parameters. The superior performance of BiDoRA verifies the effectiveness of its BLO mechanism. By training the magnitude and direction components on two distinct sub-datasets, BiDoRA enhances the flexibility of the learning process and improves learning capacity compared to DoRA, resulting in a performance boost.

We also present an experiment on the GLUE benchmark with the RoBERTa-base model, on a larger, wide range of baselines, following the settings from Shen et al. (2025b) and Shen et al. (2025a) and citing their reported baseline results for reference. The results in Table 2 indicate that BiDoRA consistently outperforms all baselines, including DoRA, across these diverse NLU tasks, demonstrating its robust generalization capability.

Table 2: Performance of various fine-tuning methods on the GLUE benchmark for the RoBERTa-base model. The best ones are highlighted by **bold** and the second ones are highlighted by *italic*.

| Model | SST-2 | MRPC | CoLA | QNLI | RTE | STS-B | Avg. |
|---|---|---|---|---|---|---|---|
| FT | 94.8 | 90.2 | 63.6 | 92.8 | 78.7 | *91.2* | 85.22 |
| BitFit | 93.7 | **92.7** | 62.0 | 91.8 | 81.5 | 90.8 | 85.42 |
| Adpt$^D$ | 94.7 | 88.4 | 62.6 | 93.0 | 75.9 | 90.3 | 84.15 |
| LoRA | *95.1* | 89.7 | 63.4 | *93.3* | 78.4 | **91.5** | 85.23 |
| AdaLoRA | 94.5 | 88.7 | 62.0 | 93.1 | 81.0 | 90.5 | 84.97 |
| AFLoRA | 94.1 | 89.3 | 63.5 | 91.3 | 77.2 | 90.6 | 84.33 |
| LaMDA | 94.6 | 89.7 | 64.9 | 91.7 | 78.2 | 90.4 | 84.92 |
| VeRA | 94.6 | 89.5 | 65.6 | 91.8 | 78.7 | 90.7 | 85.15 |
| FourierFT | 94.2 | 90.0 | 63.8 | 92.2 | 79.1 | 90.8 | 85.02 |
| SSH | 94.1 | *91.2* | 63.6 | 92.4 | 80.5 | 90.9 | 85.46 |
| MaCP | 94.2 | 89.7 | 64.6 | 92.4 | 80.7 | 90.9 | 85.42 |
| DoRA ($r = 8$) | 94.9 | 89.9 | 63.7 | *93.3* | 78.9 | **91.5** | 85.37 |
| BiDoRA ($r = 8$) | **95.7** | 90.2 | *65.8* | **93.4** | 79.4 | 90.5 | 85.83 |
| DoRA ($r = 16$) | 94.8 | 90.4 | 65.6 | 93.1 | *81.9* | 90.7 | *86.08* |
| BiDoRA ($r = 16$) | 95.0 | 90.8 | **66.7** | *93.3* | **82.6** | 90.9 | **86.55** |

Table 3: Quantitative performance gap between training and test sets for DoRA and BiDoRA using the RoBERTa-base model. The gap is calculated as the training metric minus the test metric, where a smaller value indicates less overfitting.

| Method | SST-2 | MRPC | CoLA | QNLI | RTE | STS-B | Avg. |
|---|---|---|---|---|---|---|---|
| DoRA | 2.0 | 9.5 | 32.5 | 6.6 | 18.0 | 8.8 | 12.9 |
| BiDoRA | **1.7** | **7.0** | **23.3** | **0.2** | **14.0** | **4.7** | **8.5** |

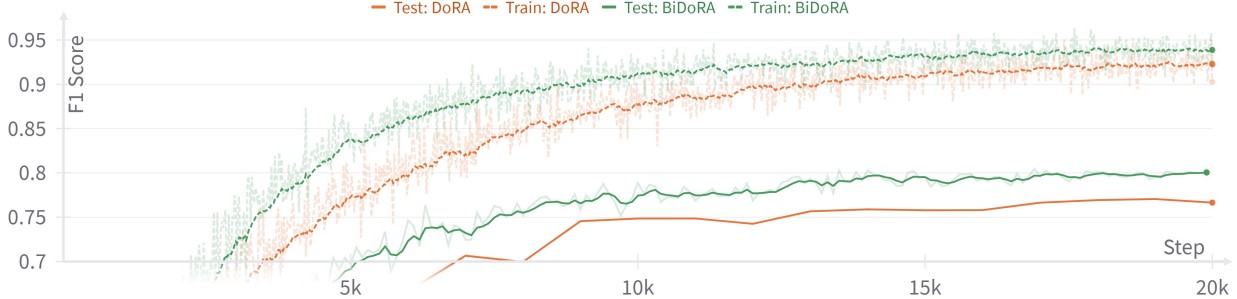

Figure 2: Training and test accuracy versus global training steps on the ModHayes split of the Reuters21578 dataset (Padmanabhan et al., 2016) when fine-tuning a RoBERTa-base model using DoRA and BiDoRA. The training and test curves for DoRA show a larger gap compared to BiDoRA, highlighting the effectiveness of our method in reducing overfitting.

**Robustness of BiDoRA towards different rank settings.** We explore the impact of different rank configurations on BiDoRA and DoRA, evaluating them with ranks of 8 and 16 in addition to the rank of 4 used in Table 1. The average accuracies reported in Table 2 demonstrate that BiDoRA consistently surpasses DoRA across all rank configurations, highlighting its resilience and superior performance regardless of the rank setting.

**Performance gap between training and testing set.** As visualized in Fig. 2, BiDoRA achieves a smaller gap between the training and test curves. Quantitatively, Table 3 presents this performance gap on the RoBERTa-base model. The training set metric is calculated as a moving average of the per-batch metric with a decay ratio of 0.99. Since BiDoRA has two training loops, its training metric is a weighted average $(0.8 \times \text{inner-loop-metric} + 0.2 \times \text{outer-loop-metric})$, based on the data split size, inner : outer = 8 : 2, in our case. The results show that the performance gap for BiDoRA is consistently lower than that of DoRA across all datasets. This suggests that DoRA is more prone to overfitting, an issue that BiDoRA effectively addresses.

## 5.3 Experiments on Extremely Small Datasets

Table 4: Fine-tuning ESM on the thermostability prediction task (Chen et al., 2023) (left), the BBP task (Dai et al., 2021) (middle), and the MIC task (Ledesma-Fernandez et al., 2023) (right). A higher value is better for all metrics except for MSE. The best results are highlighted in **bold**.

| Methods | #Params | Accuracy | Precision | Recall | F1 | #Params | Accuracy | Precision | Recall | F1 | #Params | MSE |
|---|---|---|---|---|---|---|---|---|---|---|---|---|
| FT | 652.7M | 79.8 | 81.2 | 79.8 | 78.4 | 652.9M | 89.4 | 89.9 | 89.4 | 89.4 | 652.7M | 0.2894 |
| LoRA | 1.5M | 75.9 | 78.2 | 75.9 | 75.5 | 1.9M | 86.8 | 87.7 | 86.8 | 86.7 | 1.7M | 0.3433 |
| DoRA | 1.6M | 76.9 | 78.7 | 76.9 | 76.2 | 2.0M | 89.4 | 91.3 | 89.4 | 89.3 | 1.8M | 0.2918 |
| BiDoRA | 1.6M | **78.8** | **79.1** | **78.8** | **78.2** | 2.0M | **92.1** | **93.1** | **92.1** | **92.0** | 1.8M | **0.2818** |

We conduct additional experiments on biomedical datasets, including two classification tasks—thermostability prediction (Chen et al. (2023), 936 training samples) and blood-brain barrier peptide prediction (BBP, Dai et al. (2021), 200 training samples)—and one regression task, minimum inhibitory concentration prediction (MIC, Ledesma-Fernandez et al. (2023), 3,695 training samples), which contain significantly fewer samples than standard NLP tasks.

The results are presented in Table 4. Consistent with our previous findings, BiDoRA effectively fine-tunes pre-trained models on extremely small datasets. Our method outperforms the baselines by a larger margin

Table 5: Ablation studies. We evaluate the performance of BiDoRA without retraining (w/o retraining), without BLO ($\xi = 0$), without orthogonal regularization (w/o cst.), and with retraining magnitude.

| Method | MNLI | SST-2 | MRPC | CoLA | QNLI | QQP | RTE | STS-B | Avg. |
|---|---|---|---|---|---|---|---|---|---|
| BiDoRA (retraining magnitude) | 87.0 | 94.3 | 89.1 | 60.7 | **92.7** | 91.0 | 73.4 | 89.9 | 84.8 |
| BiDoRA (w/o retraining) | 87.0 | 94.2 | 89.0 | 57.3 | 92.4 | 90.6 | 71.6 | 90.0 | 84.0 |
| BiDoRA ($\xi = 0$) | 86.9 | 94.2 | 89.0 | 59.4 | 90.8 | **91.2** | 75.9 | 90.0 | 84.7 |
| BiDoRA (w/o cst.) | 87.0 | **94.4** | 88.6 | **61.3** | **92.7** | 90.2 | 76.0 | **90.1** | 85.0 |
| BiDoRA | **87.1** | **94.4** | **89.4** | **61.3** | **92.7** | 90.6 | **76.1** | **90.1** | **85.2** |

as the dataset size decreases, confirming our previous conclusion that our method effectively combats the overfitting issue on various network architectures and diverse tasks.

## 5.4 Ablation Studies

In this section, we perform ablation studies to investigate the effectiveness of individual modules or strategies in BiDoRA. We fine-tune a RoBERTa-base model on the GLUE benchmark under different ablation settings, and the results are shown in Table 5.

**Retraining.** We test the model directly obtained from the search phase to evaluate the effectiveness of further retraining the direction component. The results show that BiDoRA outperforms BiDoRA (w/o retraining) on average, highlighting the necessity of retraining. Table 5 also validates that retraining the direction component leads to superior performance than retraining the magnitude.

**Bi-level optimization.** We set $\xi$ to zero in Algorithm 1 to assess the effectiveness of the BLO framework. This ablation setting can be interpreted as an alternative learning method where two optimization steps are carried out alternately on two different splits of the training dataset. Notably, in the alternative learning method, the updating of each component is unaware of the others, making the training less stable. In contrast, the hyper-gradient used in BLO avoids this issue by connecting the two levels in a certain way. The results show that BiDoRA outperforms BiDoRA ($\xi = 0$) on average, demonstrating the efficacy of the BLo strategy.

**Orthogonal regularization.** We examine the effectiveness of the orthogonality constraint in Eq. (3) by setting $\gamma$ to zero. Results show that BiDoRA outperforms BiDoRA (w/o cst.) on average, indicating the effectiveness of applying the orthogonality regularizer to alleviate overfitting.

## 5.5 Weight Decomposition Analysis

One important motivation of DoRA is to bridge the inherent differences between LoRA and FT. Similar to DoRA, we conduct a weight decomposition analysis on the correlation between the change of magnitudes and that of directions (detailed in Appendix D) for BiDoRA and baseline methods by fine-tuning a GPT2-medium model on the E2E dataset. As shown in Fig. 3, FT, DoRA, and BiDoRA all exhibit negative correlation values, while LoRA shows a positive correlation, consistent with the findings in Liu et al. (2024a). Notably, BiDoRA achieves a negative correlation of $-8.042$, closer to FT than DoRA's $-1.784$. This improvement is attributed to the decoupled training process of the two layers, which allows for a higher learning capacity compared to DoRA.

## 5.6 Discussion

The advantage of BiDoRA is supported by both theoretical insights and empirical evidence, as detailed as follows.

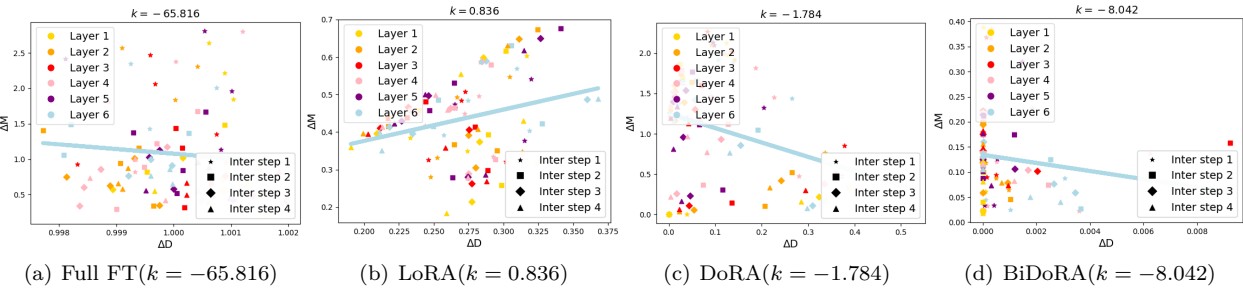

(a) Full FT($k = -65.816$)   (b) LoRA($k = 0.836$)   (c) DoRA($k = -1.784$)   (d) BiDoRA($k = -8.042$)

Figure 3: Magnitude and direction updates for (a) FT, (b) LoRA, (c) DoRA, and (d) BiDoRA of the query matrices across different layers and intermediate steps after fine-tuning the GPT2 model on the E2E dataset (Novikova et al., 2017), where $k$ denotes the correlation value. Different markers represent matrices from different training steps, with each color corresponding to a specific layer. $\mathbf{\Delta M}$ denotes the average change in weight vector magnitude, and $\mathbf{\Delta D}$ denotes the average change in direction, as formally defined in Appendix D.

**Motivation.** Theoretically, Liu et al. (2024a) showed that LoRA's training pattern tends to be coupled in terms of magnitude-direction correlation, which degrades learning capacity. Their solution was to introduce a reparameterization that decouples these components in the formulation. We build upon DoRA following their theory and further decouple magnitude and direction in terms of training dynamics. Specifically, the two components are trained in separate loops within a bilevel optimization framework, which is expected to improve performance in an intuition similar to DoRA.

**Empirical evidences.** We performed a Wilcoxon signed-rank test to compare the performance of DoRA and BiDoRA. Specifically, we used the results from Table 1. For each PEFT method, we collected 9 values (8 values from each dataset plus the average performance) from one base model. We concatenated the results from three base models (RoBERTa-base, RoBERTa-large, and DeBERTa-XXL) to obtain a list of 27 values. A comparison of these 27 values between DoRA and BiDoRA reveals that BiDoRA is significantly better than DoRA, with a p-value of $2.4 \times 10^{-4}$. This result demonstrates that BiDoRA offers a non-marginal improvement over DoRA.

Additionally, the weight decomposition analysis, including (Fig. 3 and Fig. 4), indicates that BiDoRA achieves better decoupling of the components compared to DoRA. Evaluation metrics across various tasks demonstrate the superior performance of BiDoRA, confirming that our decoupled optimization loop leads to improved outcomes.

## 6 Conclusion and Future Works

We propose BiDoRA, a novel bi-level optimization framework for PEFT of large-scale pre-trained models. By conducting weight decomposition following the DoRA approach, our method trains the two components separately in two interconnected optimization levels using different sub-datasets. In this way, BiDoRA not only decouples the learning process of the two components, resulting in a learning pattern closer to FT, but also effectively alleviates overfitting. Empirical studies on various NLP tasks demonstrate that BiDoRA outperforms DoRA and other baselines, highlighting the effectiveness of our method.

One potential limitation of BiDoRA is its training efficiency (see Appendix E) in terms of per-step cost, which could be reduced by using more advanced hyper-gradient estimators, such as SAMA (Choe et al., 2023a) or MixFlow-MG (Kemaev et al., 2025). Furthermore, while we have empirically shown that BiDoRA induces better decoupling between the magnitude and direction components (Fig. 3 and Fig. 4), a formal theoretical analysis of this property is currently lacking and serves for future work.

### Acknowledgments

P.X. acknowledges funding support from NSF IIS2405974, NSF IIS2339216, NIH R35GM157217, and NIH R21GM154171.

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

# A  Datasets and Models

Table 6: Summary of datasets used in the experiments

| Task Group | Dataset | Metrics | Train | Dev / Val | Test |
|---|---|---|---|---|---|
| Natural Language Understanding | MNLI | Accuracy | 393k | 20k | 20k |
| | SST-2 | Accuracy | 67k | 872 | 1.8k |
| | MRPC | Accuracy | 3.7k | 408 | 1.7k |
| | CoLA | Matthews Corr | 8.5k | 1k | 1k |
| | QNLI | Accuracy | 108k | 5.7k | 5.7k |
| | QQP | Accuracy | 364k | 40k | 391k |
| | RTE | Accuracy | 2.5k | 276 | 3k |
| | STS-B | Pearson Corr | 7.0k | 1.5k | 1.4k |
| Text Classification | ModApte | F1 | 8.8k | - | 3k |
| | ModHayes | F1 | 18k | - | 0.7k |
| | ModLewis | F1 | 12k | - | 5.5k |
| Natural Language Generation | E2E | BLEU, NIST, MET, ROUGE-L, CIDEr | 42k | 4.6k | - |
| Token Classification | BioNLP | Accuracy, Precision, Recall, F1 | 17k | 1.9k | 3.9k |
| | CoNLL2003 | Accuracy, Precision, Recall, F1 | 14k | 3.3k | 3.5k |
| Biomedical Experiments | Thermostability prediction | Accuracy, Precision, Recall, F1 | 3,695 | - | 924 |
| | BBP | Accuracy, Precision, Recall, F1 | 200 | - | 38 |
| | MIC | MSE | 936 | - | 104 |

In this section, we present the datasets and models used in experiments, and summarize the statistical data in Table 6.

## A.1  Natural Language Understanding

The GLUE Benchmark (Wang et al., 2019) comprises a diverse array of tasks that are widely employed for evaluation in natural language understanding. It encompasses two single-sentence classification tasks, three tasks assessing similarity and paraphrasing, and four tasks focusing on natural language inference. Specifically, it includes MNLI (MultiNLI, Williams et al. (2018)), SST-2 (Stanford Sentiment Treebank, Socher et al. (2013)), MRPC (Microsoft Research Paraphrase Corpus, Dolan & Brockett (2005)), CoLA (Corpus of Linguistic Acceptability, Warstadt et al. (2019)), QNLI (Question NLI, Rajpurkar et al. (2018)), QQP (Quora Question Pairs, Wang et al. (2017)), RTE (Recognizing Textual Entailment, Dagan et al. (2005)), and STS-B (Semantic Textual Similarity Benchmark, Cer et al. (2017)). We summarize the statistical data for all datasets within the GLUE Benchmark in Table 6. Following existing practices, the development set is used in GLUE as the test data since the actual test set is not publicly available. We report the overall (matched and mismatched) accuracy for MNLI, Matthew's correlation for CoLA, Pearson correlation for STS-B, and accuracy for the other tasks.

The Reuters-21578 (Padmanabhan et al., 2016) dataset is one of the most widely used data collections for text categorization research. It was collected from the Reuters financial newswire service in 1987 and is used for text classification and natural language processing tasks. Three splits are available: ModApte,

ModHayes, and ModLewis. These documents cover various topics, such as politics, economics, and sports. F1 score is used as the evaluation metric across all three splits.

## A.2 Natural Language Generation

In our experiments on natural language generation, we use the E2E (Novikova et al., 2017) dataset, which was initially introduced as a dataset for training end-to-end, data-driven natural language generation systems. Multiple references can be associated with each source table used as input. Each sample input $(x, y)$ consists of a series of slot-value pairs accompanied by an associated natural language reference text. The E2E dataset comprises approximately 42k training examples, $4,600$ validation examples, and $4,600$ test examples from the restaurant domain.

We utilize the following five evaluation metrics: BLEU (Papineni et al., 2002), NIST (Lin & Och, 2004), METEOR (Banerjee & Lavie, 2005), ROUGE-L (Lin, 2004), and CIDEr (Vedantam et al., 2015).

## A.3 Token Classification

For token classification, we fine-tune the RoBERTa-base and RoBERTa-large models on the BioNLP dataset (Collier et al., 2004) and the CoNLL2003 dataset (Tjong Kim Sang, 2002). BioNLP (Collier et al., 2004) is a Named Entity Recognition dataset that contains biological entities such as DNA, RNA, and protein. It is essentially a token classification task where we want to classify each entity in the sequence. CoNLL-2003 (Tjong Kim Sang, 2002) focuses on language-independent named entity recognition. It concentrates on four types of named entities: persons, locations, organizations, and miscellaneous entities that do not belong to the previous three groups. Accuracy, precision, recall, and F1 score are used as evaluation metrics.

## A.4 Biomedical Experiments

The ESM (Evolutionary Scale Modeling, Rives et al. (2021)) model is a transformer-based protein language model designed for protein sequence analysis, leveraging the transformer architecture to capture evolutionary patterns. We fine-tune the ESM model using the Protein Aligner checkpoint (Zhang et al., 2024a) on two classification tasks—thermostability prediction (Chen et al. (2023), 936 training samples) and blood-brain barrier peptide prediction (BBP, Dai et al. (2021), 200 training samples)—and one regression task, minimum inhibitory concentration prediction (MIC, Ledesma-Fernandez et al. (2023), 3,695 training samples). Notably, protein analysis datasets are typically much smaller than those in NLP, in which case the large pre-trained models are prone to overfitting, even when using PEFT methods. The trainable parameters (on the order of millions) are significantly overparameterized compared to the available samples (thousands or even hundreds), highlighting the need for our overfitting-resilient counterpart.

# B Experimental Settings

In this section, we provide detailed experimental settings. We maintain consistent configurations across experiments, including LoRA rank, LoRA $\alpha$, batch size, maximum sequence length, and optimizer, to ensure a fair comparison. For results other than Table 2, we do not include the bias term in PEFT linear layers. The hyperparameter tuning for our method is straightforward and convenient.

## B.1 RoBERTa

We summarize the experimental settings for the GLUE benchmark (Table 1) and for the Reuters21578 dataset and token classification (Table 13) tasks in Table 7.

## B.2 GPT-2

We summarize the experimental settings for the GPT-2 experiments (Table 12) in Table 8. The experimental configuration, particularly during the inference stage, follows the approach described by Hu et al. (2022b).

Table 7: The hyperparameters used for RoBERTa on the GLUE benchmark (Wang et al., 2019), Reuters21578 dataset (Padmanabhan et al., 2016), BioNLP dataset (Collier et al., 2004), and CoNLL2003 dataset (Tjong Kim Sang, 2002).

| | Settings | MNLI | SST-2 | MRPC | CoLA | QNLI | QQP | RTE | STS-B | ModApte | ModHayes | ModLewis | BioNLP | CoNLL2003 |
|---|---|---|---|---|---|---|---|---|---|---|---|---|---|---|
| | Optimizer | | | | | | | | AdamW | | | | | |
| | Warmup Ratio | | | | | | | | 0.06 | | | | | |
| | Scheduler | | | | | | | | Linear | | | | | |
| | LoRA rank | | | | | | | | rank = 4 | | | | | |
| | LoRA $\alpha$ | | | | | | | | 8 | | | | | |
| RoBERTa-base | Total batch size | | | | | | | | 32 | | | | | |
| | Global steps | 20k | 12k | 25k | 20k | 15k | 20k | 15k | 12k | 20k | 20k | 20k | 12k | 12k |
| | Lower learning rate | 5e-5 | 1e-5 | 2e-5 | 5e-5 | 2e-5 | 5e-5 | 1e-5 | 1e-5 | 3e-5 | 3e-5 | 3e-5 | 1e-5 | 2e-5 |
| | Upper learning rate | 5e-5 | 1e-5 | 2e-5 | 5e-5 | 2e-5 | 5e-5 | 1e-5 | 1e-5 | 3e-5 | 3e-5 | 3e-5 | 1e-5 | 2e-5 |
| | Lower weight decay | 0.1 | 0.1 | 0.1 | 0.1 | 0.1 | 0.1 | 0.1 | 0.1 | 0.1 | 0.1 | 0.1 | 0.1 | 0.2 |
| | Upper weight decay | 0.1 | 0.1 | 0.1 | 0.1 | 0 | 0.1 | 0.1 | 0.01 | 0.1 | 0.1 | 0.1 | 0.1 | 0.1 |
| | Max Seq Length | | | | | | | | 512 | | | | | |
| | Regularization Coefficient | 1e-5 | 1e-5 | 1e-5 | 1e-5 | 1e-5 | 1e-5 | 1e-5 | 1e-5 | 0 | 1e-5 | 0 | 1e-5 | 0 |
| RoBERTa-large | Total batch size | | | | | | | | 32 | | | | | |
| | Global steps | 50k | 20k | 30k | 20k | 60k | 40k | 15k | 10k | 20k | 20k | 20k | 12k | 15k |
| | Lower learning rate | 1e-5 | 1e-5 | 1e-5 | 1e-5 | 1e-5 | 1e-5 | 1e-5 | 1e-5 | 1e-5 | 1e-5 | 1e-5 | 2e-5 | 1e-5 |
| | Upper learning rate | 1e-5 | 1e-5 | 1e-5 | 1e-5 | 1e-5 | 1e-5 | 1e-5 | 1e-5 | 1e-5 | 1e-5 | 1e-5 | 2e-5 | 1e-5 |
| | Lower weight decay | 0.5 | 0.5 | 0 | 0.2 | 0.5 | 0.5 | 0.5 | 0.5 | 0.2 | 0.1 | 0.2 | 0.02 | 0.1 |
| | Upper weight decay | 0.5 | 0.05 | 0 | 0.2 | 0.5 | 0.5 | 0.1 | 0.5 | 0.1 | 0.1 | 0.1 | 0.02 | 0.1 |
| | Max Seq Length | | | | | | | | 128 | | | | | |
| | Regularization Coefficient | 0 | 0 | 1e-5 | 1e-5 | 0 | 1e-5 | 0 | 1e-5 | 0 | 1e-5 | 0 | 0 | 1e-5 |

Table 8: The hyperparameters we used for GPT-2 on the E2E NLG benchmark (Novikova et al., 2017).

| Settings | Training |
|---|---|
| Optimizer | AdamW |
| Warmup Ratio | 0.06 |
| Scheduler | Linear |
| LoRA rank | $\text{rank}_a = \text{rank}_u = 4$ |
| LoRA $\alpha$ | 32 |
| Label Smooth | 0.1 |
| Lower learning rate | 1e-3 |
| Upper learning rate | 1e-4 |
| Lower weight decay | 1 |
| Upper weight decay | 1 |
| Max Seq Length | 512 |
| Regularization Coefficient | 1e-5 |
| Settings | Inference |
| Beam Size | 10 |
| Length Penalty | 0.9 |
| no repeat ngram size | 4 |

## C   Baselines in Experiments

Here, we provide a brief introduction to compare baselines in all our experiments.

- **Full Fine-Tuning (FT):** The entire model is fine-tuned, with updates to all parameters.

- **Adapter Tuning (Houlsby et al., 2019; Lin et al., 2020; Rücklé et al., 2021; Pfeiffer et al., 2021):** Methods that introduce adapter layers between the self-attention and MLP modules for parameter-efficient tuning.

- **LoRA (Hu et al., 2022a):** A method that estimates weight updates via low-rank matrices.

- **AdaLoRA (Zhang et al., 2023):** An extension of LoRA that dynamically reallocates the parameter budget based on importance scores.

- **DoRA (Liu et al., 2024a):** Decomposes pretrained weights into magnitude and direction, using LoRA for efficient directional updates.

- **VeRA (Kopiczko et al., 2024):** Employs a single pair of low-rank matrices across all layers to reduce trainable parameters.

- **FourierFT (Gao et al., 2024):** Fine-tunes models by learning a subset of spectral coefficients in the Fourier domain.

- **AFLoRA (Liu et al., 2024b):** Freezes low-rank adaptation parameters using a learned score, reducing trainable parameters while maintaining performance.

- **LaMDA (Azizi et al., 2024):** Fine-tunes large models via spectrally decomposed low-dimensional adaptation.

- **SSH (Shen et al., 2025b):** Fine-tunes large models after transforming weight matrices with the discrete Hartley transformation (DHT).

- **MaCP (Shen et al., 2025a):** Fine-tunes large models by projecting the low-rank adaptation weight change into the discrete cosine space.

## D  Weight Decomposition Analysis

We provide a brief review of the weight decomposition analysis proposed in Liu et al. (2024a). Define the weight decomposition of a weight matrix $\mathbf{W} \in \mathbb{R}^{d \times k}$ (e.g., query matrix in an attention layer) as $\mathbf{W} = \mathbf{m} \frac{\mathbf{V}}{\|\mathbf{V}\|_c} = \|\mathbf{W}\|_c \frac{\mathbf{W}}{\|\mathbf{W}\|_c}$, where $\mathbf{m} \in \mathbb{R}^{1 \times k}$ is the magnitude vector, and $\mathbf{V} \in \mathbb{R}^{d \times k}$ is the directional matrix, with $\|\cdot\|_c$ representing the vector-wise norm of a matrix across each column. This decomposition ensures that each column of $\mathbf{V}/\|\mathbf{V}\|_c$ remains a unit vector, and the corresponding scalar in $\mathbf{m}$ defines the magnitude of each vector. Liu et al. (2024a) examine the magnitude and directional variations between $\mathbf{W_0}$ and $\mathbf{W}_{\mathrm{FT}}$, defined as $\Delta \mathbf{M}_{\mathrm{FT}}^t = \frac{\sum_{n=1}^{k} |\mathbf{m}_{\mathrm{FT}}^{n,t} - \mathbf{m}_0^n|}{k}$ and $\Delta \mathbf{D}_{\mathrm{FT}}^t = \frac{\sum_{n=1}^{k}(1 - \cos(\mathbf{V}_{\mathrm{FT}}^{n,t}, \mathbf{W_0}^n))}{k}$. Here, $\Delta \mathbf{M}_{\mathrm{FT}}^t$ and $\Delta \mathbf{D}_{\mathrm{FT}}^t$ represent the magnitude and direction differences between $W_0$ and $W_{\mathrm{FT}}$ at the $t$-th training step, respectively, with $\cos(\cdot, \cdot)$ denoting cosine similarity. $m_{\mathrm{FT}}^{n,t}$ and $m_0^n$ are the $n^{\mathrm{th}}$ scalars in their respective magnitude vectors, while $\mathbf{V}_{\mathrm{FT}}^{n,t}$ and $\mathbf{W_0}^n$ are the $n^{\mathrm{th}}$ columns in $\mathbf{V}_{\mathrm{FT}}^t$ and $\mathbf{W_0}$. Intuitively, a consistent positive slope trend across all the intermediate steps implies a difficulty in concurrent learning of both magnitude and direction, suggesting that slight directional changes are challenging to execute alongside more significant magnitude alterations. In contrast, a relatively negative slope signifies a more varied learning pattern, with a more pronounced negative correlation indicating a larger learning capacity.

Complementary to Fig. 3 in the main paper on the query matrix, we provide additional results of weight decomposition analysis in Fig. 4 on the value matrix to complement the findings in Section 5.5. We can draw two key observations from Fig. 4: 1) Consistent with the results in Liu et al. (2024a), both FT and DoRA exhibit negative correlation values of $-49.279$ and $-5.485$, respectively, while LoRA shows a positive correlation with a value of $2.503$. 2) BiDoRA achieves a negative correlation value of $-10.547$, indicating closer alignment with FT compared to DoRA. The analysis of how BiDoRA achieves this improvement is similar to that discussed in Section 5.5.

## E  Training Cost

Table 9 compares the training efficiency of LoRA, DoRA, and BiDoRA on the GLUE benchmark using the RoBERTa-base model. The table details the total training steps required for convergence and the per-step computational cost, which is normalized relative to LoRA for reference. For a fair comparison, all methods

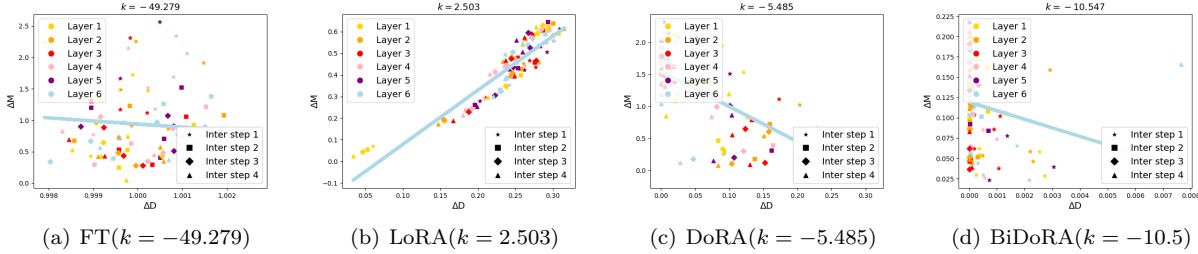

(a) FT($k = -49.279$)  (b) LoRA($k = 2.503$)  (c) DoRA($k = -5.485$)  (d) BiDoRA($k = -10.5$)

Figure 4: **Magnitude and direction updates** for (a) FT, (b) LoRA, (c) DoRA, and (d) BiDoRA of the value matrices across different layers and intermediate steps after fine-tuning the GPT2 model on the E2E dataset (Novikova et al., 2017). Different markers represent matrices from different training steps, while different colors indicate matrices from each layer. The values of negative correlation are shown at the top, denoted by $k$.

Table 9: Average training time cost on the GLUE benchmark (Wang et al., 2019).

| Method | LoRA | DoRA | BiDoRA |
|---|---|---|---|
| Per-step cost | ×1 | ×1.36 | ×3.54 |
| Total steps | 27.45k | 27.45k | 17.37k |
| Total time | ×1 | ×1.36 | ×2.24 |

were benchmarked on a single NVIDIA A100 GPU. The results show that BiDoRA converges in fewer steps than LoRA and DoRA, while the per-step cost for BiDoRA is modestly higher, as its BLO process requires iterative updates between the two levels and the computation of hypergradients. The total training time for BiDoRA is approximately 1.64 times that of DoRA, a training cost that remains comparable to the baselines. Given BiDoRA's superior performance across various tasks, we argue that this slight increase in computational cost is an acceptable trade-off, underscoring our method's practicality.

## F   The Role of Hyperparameter

The hyperparameter tuning for BiDoRA is simple, convenient, and straightforward. We further conducted experiments regarding the dataset partition of $\mathcal{D}_{tr}$ and $\mathcal{D}_{val}$ to provide insights into its role in BiDoRA. The dataset partition helps maintain the balance of inner/outer optimization by assigning different portions of data. The direction component has more trainable parameters, so it is reasonable to use more data for training the lower level while using the remaining data for training magnitudes. As shown in Table 10, we varied the inner-level dataset $\mathcal{D}_{tr}$ partition from 0.6 to 1.0 with 0.1 intervals and experimented with RoBERTa-base on three splits of the Reuters21578 dataset to examine its influence.

The results indicate that both extreme cases are negative to the overall performance. When the inner partition is too small ($\leq 0.6$), directions are not well-trained, and when the inner partition is 1.0, magnitudes are not trained at all, leading to a significant performance drop. These findings demonstrate that BLO is effective in the sense that both levels are necessary for enhancing performance. Although tuning the partition ratio may further improve overall performance, we maintain a consistent data partition of $8 : 2$ in all the experiments for simplicity. A fixed configuration of data partition already consistently yields superior performance of BiDoRA, demonstrating that our method is robust to this hyperparameter within a certain range.

Table 10: Experiment results on different data partitions of BiDoRA.

| Partition | ModApte | ModHayes | ModLewis |
|:---:|:---:|:---:|:---:|
| 0.6 | 85.32 | 79.76 | 77.69 |
| 0.7 | 85.32 | **80.01** | **77.74** |
| 0.8 | **85.34** | 79.93 | 77.63 |
| 0.9 | 85.27 | 79.85 | 77.64 |
| 1.0 | 85.23 | 79.59 | 77.42 |

Table 11: Experiment results on different weight decay values and different dropout rates of DoRA.

| Method | CoLA | MRPC | RTE |
|:---|:---:|:---:|:---:|
| DoRA (weight decay = 0) | 59.3 | 88.7 | 72.9 |
| DoRA (weight decay = 0.05) | 60.1 | 89.2 | 73.3 |
| DoRA (weight decay = 0.1) | 60.5 | 89.2 | 73.2 |
| DoRA (weight decay = 0.2) | 60.3 | 89.0 | 73.2 |
| DoRA (dropout rate = 0) | 59.2 | 89.2 | 72.9 |
| DoRA (dropout rate = 0.1) | 60.2 | 88.9 | 71.4 |
| DoRA (dropout rate = 0.2) | 55.1 | 87.8 | 64.2 |
| BiDoRA | **61.3** | **89.4** | **76.0** |

## G Comparison with Other General Methods for Addressing Overfitting

There are some common experimental settings that may be used to reduce overfitting. For DoRA, two promising methods are increasing weight decay and adopting a more aggressive dropout rate. We conducted experiments on these two methods separately. We kept hyperparameters that have been well-tuned in DoRA and can achieve optimal results while only tuning the weight decay value. Similarly, we tune the dropout rate of DoRA while keeping the weight decay value to be optimized. We conducted experiments on RoBERTa-base on three datasets. The results are presented in Table 11.

We can draw the observation that neither of these approaches effectively addresses overfitting issues or enhances the model's generalization ability. On the other hand, BiDoRA exploits the specific magnitude-direction structure of DoRA and the strategy of training the two distinct components on separate splits of the dataset. An advantage of our methodology is that it can be easily combined with other general-purpose overfitting-alleviating strategies since BiDoRA does not modify the original DoRA architecture.

## H Additional Experiments

### H.1 Experiments on Natural Language Generation Tasks

In this section, we evaluate BiDoRA's performance on the NLG task. Table 12 presents the results of fine-tuning a GPT-2 model on the E2E dataset with baseline PEFT methods and BiDoRA. The results show that BiDoRA achieves the best performance across all five evaluation metrics, demonstrating the superiority of BiDoRA in fine-tuning pre-trained models for NLG tasks.

### H.2 Experiments on Token Classification

The effectiveness of BiDoRA can also be observed in Table 13, which reports the results of token classification tasks. Unlike the NLU tasks discussed in the previous section, which involve classifying entire sentences and focusing on capturing global semantics, token classification requires classifying each token within a sentence, highlighting the importance of capturing local context. On the BioNLP dataset, BiDoRA consistently

Table 12: Performance of BiDoRA and baseline methods for fine-tuning GPT2-medium on the E2E dataset (Novikova et al., 2017). A higher value is better for all metrics. The best results are shown in **bold**.

| Method | #Params | BLEU | NIST | MET | ROUGE-L | CIDEr |
|--------|---------|------|------|-----|---------|-------|
| FT | 354.9M | 68.0 | 8.61 | 46.1 | 69.0 | 2.38 |
| Adapter | 11.1M | 67.0 | 8.50 | 45.2 | 66.9 | 2.31 |
| LoRA | 0.39M | 67.1 | 8.54 | 45.7 | 68.0 | 2.33 |
| DoRA | 0.39M | 67.0 | 8.48 | 45.4 | 70.1 | 2.33 |
| BiDoRA | 0.39M | **69.0** | **8.72** | **46.2** | **70.9** | **2.44** |

outperforms baseline methods by a large margin in terms of F1 score. On the CoNLL2003 dataset, BiDoRA either outperforms or matches all baseline methods across all metrics. Consistent with our previous findings, BiDoRA effectively fine-tunes pre-trained models for token classification tasks.

### H.3 More Experiments on Natural Language Understanding Tasks

Table 13 presents the results of fine-tuning RoBERTa models on the Reuters21578 datasets, a text classification task, where BiDoRA outperforms all baseline methods by an even larger margin. Notably, BiDoRA achieves performance comparable to or even better than full fine-tuning, providing further evidence of its superiority.

Table 13: RoBERTa$_{base/large}$ ($R_{b/l}$) with different fine-tuning methods on the Reuters21578 (Padmanabhan et al., 2016), BioNLP (Collier et al., 2004), and CoNLL2003 (Tjong Kim Sang, 2002) benchmarks. A higher value is better for all metrics. The best results are shown in **bold**.

| Method | #Params | Reuters21578 ModApte | ModHayes | ModLewis | BioNLP Accuracy | Precision | Recall | F1 | CoNLL2003 Accuracy | Precision | Recall | F1 |
|--------|---------|---------|----------|----------|----------|-----------|--------|-----|----------|-----------|--------|-----|
| $R_b$(FT) | 125.0M | 85.4 | 77.6 | 77.1 | 93.9 | 69.0 | 78.9 | 73.6 | 99.3 | 95.7 | 96.3 | 96.0 |
| $R_b$(Adapter) | 0.9M | **85.3** | 77.5 | 76.8 | 93.9 | 69.1 | 78.8 | 73.7 | **99.3** | 95.7 | 96.4 | 96.0 |
| $R_b$(LoRA) | 0.15M | 84.7 | 74.3 | 74.7 | 93.9 | 69.0 | 78.8 | 73.6 | **99.3** | 95.4 | 96.3 | 95.8 |
| $R_b$(DoRA) | 0.17M | 84.8 | 78.2 | 76.6 | **94.0** | 69.2 | **79.1** | 73.8 | **99.3** | 95.3 | 96.2 | 95.8 |
| $R_b$(BiDoRA) | 0.17M | **85.3** | **79.9** | **77.6** | 93.9 | **71.2** | 78.6 | **74.7** | **99.3** | 95.9 | **96.5** | **96.2** |
| $R_l$(FT) | 355.0M | 84.8 | 77.5 | 76.6 | 94.0 | 69.4 | 79.6 | 74.1 | 99.4 | 96.2 | 97.0 | 96.6 |
| $R_l$(Adapter) | 0.44M | 84.8 | 77.9 | 76.7 | **94.0** | 69.4 | 79.7 | 74.2 | **99.4** | 96.1 | 97.0 | 96.6 |
| $R_l$(LoRA) | 0.39M | 84.7 | 77.7 | 76.7 | 93.9 | 69.2 | 79.3 | 73.9 | **99.4** | 96.2 | 97.0 | 96.6 |
| $R_l$(DoRA) | 0.39M | 84.8 | 77.4 | 76.7 | **94.0** | 69.4 | **79.7** | 74.2 | **99.4** | 96.2 | **97.1** | 96.6 |
| $R_l$(BiDoRA) | 0.39M | **84.9** | **78.9** | **77.3** | **94.0** | **71.3** | 79.3 | **75.1** | **99.4** | **96.4** | **97.1** | **96.7** |

## I Evidence on Orthogonality of Incremental Matrix

To verify that the orthogonal regularization (OR) proposed in Section 4.2 encourages the columns of the direction matrix to be orthogonal, we visualize the normalized eigenvalues of the matrix in Fig. 5. The results show that, compared to methods without OR (i.e., DoRA and BiDoRA w/o cst.), BiDoRA with OR produces eigenvalues that are more closely aligned with those of a purely orthogonal matrix, where all eigenvalues would be one. This effect holds for both the query and value matrices and verifies the effectiveness of the OR constraint.

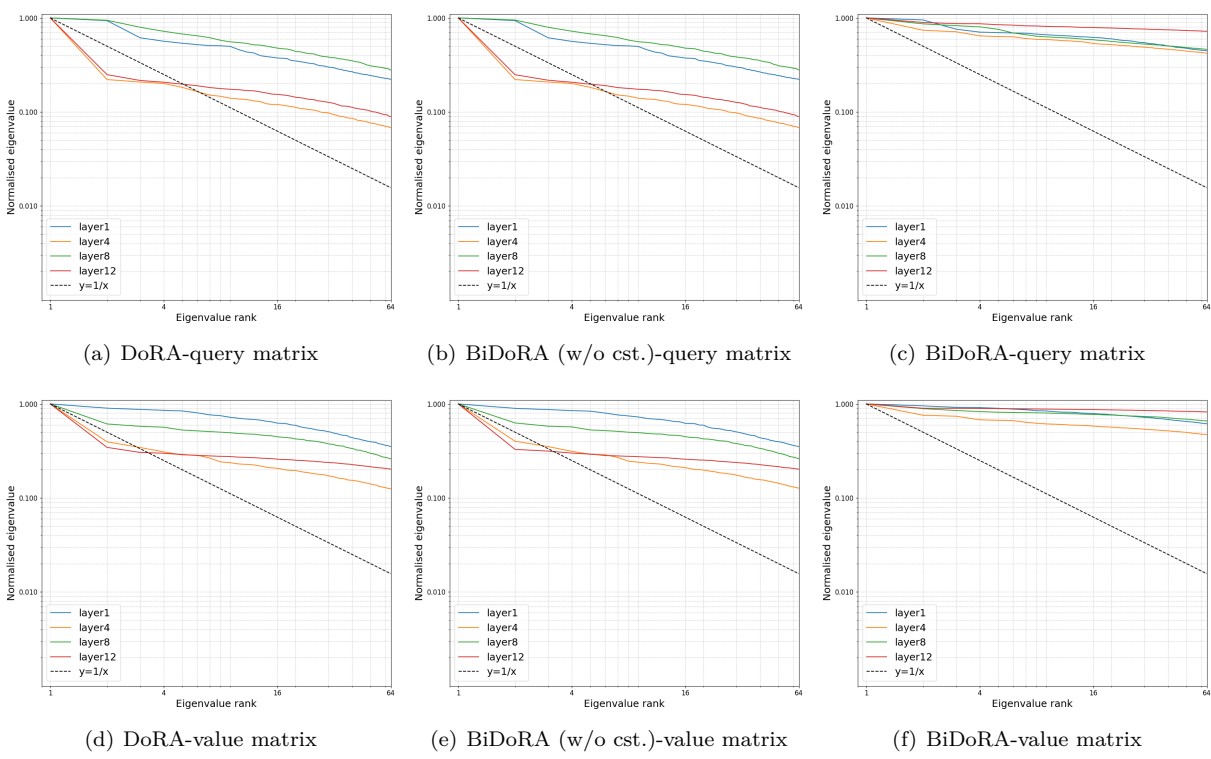

Figure 5: Eigenspectra of the direction matrix for query (top) and value (bottom) matrices across different layers. The figure compares three fine-tuning methods: BiDoRA, BiDoRA without orthogonal regularization (w/o cst.), and DoRA. Both axes are on a log scale, and only the 64 largest eigenvalues are shown for visualization clarity. Experiments were conducted on the CoLA dataset (Warstadt et al., 2019) with the RoBERTa-base model.

