# OpenReview forum: "BiDoRA: Bi-level Optimization-Based Weight-Decomposed Low-Rank Adaptation"
_TMLR — Accepted by TMLR_

### Review · Reviewer_nEhv · 2025-06-07

**Summary Of Contributions:**

This paper focuses on parameter-efficient fine-tuning (PEFT), which is a fundamental problem to harness the foundation model to various down-stream tasks.

The authors proposes a variation of weighted decomposed low-rank adaptation (DoRA), which leverages the bi-level optimization to mitigate the risk of overfitting.

Experiments on multiple downstream tasks show its effectiveness over the baseline.

**Audience:**

Yes

**Broader Impact Concerns:**

The reviewer does not envision negative societal or broader negative impact on this work.

**Claims And Evidence:**

Yes

**Requested Changes:**

Please refer to each point of weakness than has been mentioned, which the reviewer has been attached as follows.

Major concerns:

- Sec.5.6, ablation study. More visual analysis and empirical evidence is needed to support the effectiveness of the orthogonal regularization. This could be, for example, a feature space comparison, or a distance measurement comparison, with and without the use of this operation.

- Sec.5.7, when discussing the weight decomposition, there is a lack of the visualized weigh matrix. It would be much more convincing to compare the weight matrix with and without the use of the proposed method.

- The abstract needs a significant improvement.

(1) The background of this task is too long, occupying 60% of its content. Please make it more concise.

(2) The abstract should mention the numerical improvement.

- The variation of LoRA has been extensively studied in the past three years. This paper lacks a discussion, and if necessary a comparison, with some more recent state-of-the-art LoRA methods, for example:

[1] Shen, Yixian, et al. "SSH: Sparse Spectrum Adaptation via Discrete Hartley Transformation." NAACL, 2025.

[2] Hayou, Soufiane, Nikhil Ghosh, and Bin Yu. "LoRA+: Efficient Low Rank Adaptation of Large Models." International Conference on Machine Learning. PMLR, 2024.

[3] Shen, Yixian, et al. "MaCP: Minimal yet Mighty Adaptation via Hierarchical Cosine Projection." ACL, 2025.

Please enrich it for discussion and if necessary comparison.

- There are still multiple issues on the mathematical notations and equations. For example:

(1) All the tensors and matrices should be in bold to distinguish them from the scalar.

(2) ‘|| · ||_c represents the vector-wise norm of a matrix compute …’. Which specific norm operation do you use? Please clarify.

(3) For the loss function in Sec.4.3. It may be better to keep consistent to use $\mathcal{L}$ instead of $\mathcal{C}$.

(4) An equation in the optimization algorithm does not have the equation number.

- In the related work section, the adapter based PEFT methods are too old. The authors are suggested to discuss more recent PEFT methods in 2023-2024, and give their due credit to the adapter field. For example:

[4] Bi, Qi, et al. "Learning frequency-adapted vision foundation model for domain generalized semantic segmentation." Advances in Neural Information Processing Systems 37 (2024): 94047-94072.

[5] Xu, Mengde, et al. "Side adapter network for open-vocabulary semantic segmentation." Proceedings of the IEEE/CVF conference on computer vision and pattern recognition. 2023.

[6] Yi, Jingjun, et al. "Learning spectral-decomposited tokens for domain generalized semantic segmentation." Proceedings of the 32nd ACM International Conference on Multimedia. 2024.

- Caption of Fig.3. Please define $\Delta D$ and $\Delta M$ in the caption.

- Can some failure cases and limitation be discussed at the end of this paper?

- In the Section E of the appendix, please explicitly mention which specific type of the hardware the authors use for the experiments.

Minor comments for improvement:

- In all the tables’ caption, when there is a dataset, please cite it.

- The content in the top slide of Fig.1 should be made along the same height.

- The caption of Fig.2. Please cite the dataset.

- There are still multiple grammar issues in this paper. The authors need a throughout check. For example:

(1) ‘…, where architecture and sub-network are learned …’

(2) ‘Furthermore, the asynchronous gradient update steps at the two optimization levels in BiDoRA facilitate better decoupling of two components’

**Strengths And Weaknesses:**

Strength:

+ Overall this work is moderately novel for publication.

+ This paper is generally easy-to-follow with enough clarity.

Weakness: However, there are still multiple issues in terms of experimental validation, recent state-of-the-art works and presentation. Therefore, a major revision is needed before possible publication. Please refer to the specific comments as follows.

Major concerns:

- Sec.5.6, ablation study. More visual analysis and empirical evidence is needed to support the effectiveness of the orthogonal regularization. This could be, for example, a feature space comparison, or a distance measurement comparison, with and without the use of this operation.

- Sec.5.7, when discussing the weight decomposition, there is a lack of the visualized weigh matrix. It would be much more convincing to compare the weight matrix with and without the use of the proposed method.

- The abstract needs a significant improvement.

(1) The background of this task is too long, occupying 60% of its content. Please make it more concise.

(2) The abstract should mention the numerical improvement.

- The variation of LoRA has been extensively studied in the past three years. This paper lacks a discussion, and if necessary a comparison, with some more recent state-of-the-art LoRA methods, for example:

[1] Shen, Yixian, et al. "SSH: Sparse Spectrum Adaptation via Discrete Hartley Transformation." NAACL, 2025.

[2] Hayou, Soufiane, Nikhil Ghosh, and Bin Yu. "LoRA+: Efficient Low Rank Adaptation of Large Models." International Conference on Machine Learning. PMLR, 2024.

[3] Shen, Yixian, et al. "MaCP: Minimal yet Mighty Adaptation via Hierarchical Cosine Projection." ACL, 2025.

Please enrich it for discussion and if necessary comparison.

- There are still multiple issues on the mathematical notations and equations. For example:

(1) All the tensors and matrices should be in bold to distinguish them from the scalar.

(2) ‘|| · ||_c represents the vector-wise norm of a matrix compute …’. Which specific norm operation do you use? Please clarify.

(3) For the loss function in Sec.4.3. It may be better to keep consistent to use $\mathcal{L}$ instead of $\mathcal{C}$.

(4) An equation in the optimization algorithm does not have the equation number.

- In the related work section, the adapter based PEFT methods are too old. The authors are suggested to discuss more recent PEFT methods in 2023-2024, and give their due credit to the adapter field. For example:

[4] Bi, Qi, et al. "Learning frequency-adapted vision foundation model for domain generalized semantic segmentation." Advances in Neural Information Processing Systems 37 (2024): 94047-94072.

[5] Xu, Mengde, et al. "Side adapter network for open-vocabulary semantic segmentation." Proceedings of the IEEE/CVF conference on computer vision and pattern recognition. 2023.

[6] Yi, Jingjun, et al. "Learning spectral-decomposited tokens for domain generalized semantic segmentation." Proceedings of the 32nd ACM International Conference on Multimedia. 2024.

- Caption of Fig.3. Please define $\Delta D$ and $\Delta M$ in the caption.

- Can some failure cases and limitation be discussed at the end of this paper?

- In the Section E of the appendix, please explicitly mention which specific type of the hardware the authors use for the experiments.

Minor comments for improvement:

- In all the tables’ caption, when there is a dataset, please cite it.

- The content in the top slide of Fig.1 should be made along the same height.

- The caption of Fig.2. Please cite the dataset.

- There are still multiple grammar issues in this paper. The authors need a throughout check. For example:

(1) ‘…, where architecture and sub-network are learned …’

(2) ‘Furthermore, the asynchronous gradient update steps at the two optimization levels in BiDoRA facilitate better decoupling of two components’

---

> ### Author Response · Authors · 2025-06-27
> **Official Rebuttal**
>
> We appreciate your constructive feedback very much. We provide our response to your review as follows.
>
> > Sec.5.6, ablation study. More visual analysis and empirical evidence is needed to support the effectiveness of the orthogonal regularization.
>
> We have now provided additional visual evidence in Appendix I.
> We calculate the eigenvalues of the direction matrices and compare them with those of a purely orthogonal matrix,
> whose eigenvalues are all one.
> We plot the largest 64 eigenvalues for three methods: DoRA, BiDoRA without orthogonal regularization (OR), and BiDoRA with OR.
> Notably, the eigenvalue pattern of BiDoRA with OR is closely aligned with that of an orthogonal matrix,
> which demonstrates the effectiveness of the OR constraint.
>
> > Sec.5.7, when discussing the weight decomposition, there is a lack of the visualized weigh matrix.
>
> We apologize for the confusion.
> The goal of the weight decomposition analysis is to compare the *changes* in the weight matrix (both magnitude and direction) during fine-tuning,
> rather than visualizing the weight matrix itself.
> Therefore, in Section 5.7 and Figures 3-4, we compare the weight matrices with and without our method *in terms of their changes over time*.
> The rationale for comparing these changes is detailed in Appendix D;
> we directly follow the experimental protocol from the original DoRA paper for the weight decomposition analysis.
>
> > The abstract needs a significant improvement. / The abstract should mention the numerical improvement.
>
> Thank you for pointing this out.
> We have rewritten the abstract to improve its presentation, including numerical results and a clearer motivation.
>
> > The variation of LoRA has been extensively studied in the past three years. This paper lacks a discussion.
>
> Thank you for raising this crucial issue.
> To address your concern, we have added an extensive comparison with a wide range of recent PEFT methods in Appendix H.1.
> In addition to the methods compared in the main paper, we now include comparisons with the following:
>
> - AdaLoRA: Qingru Zhang, Minshuo Chen, Alexander Bukharin, Pengcheng He, Yu Cheng, Weizhu Chen, and Tuo Zhao. Adaptive budget allocation for parameter-efficient fine-tuning. In The Eleventh International Conference on Learning Representations.
> - VeRA: Kopiczko, Dawid Jan, Tijmen Blankevoort, and Yuki M. Asano. "VeRA: Vector-based Random Matrix Adaptation." The Twelfth International Conference on Learning Representations.
> - FourierFT: Gao, Ziqi, et al. "Parameter-efficient fine-tuning with discrete fourier transform." Proceedings of the 41st International Conference on Machine Learning. 2024.
> - AFLoRA: Liu, Zeyu, et al. "AFLoRA: Adaptive Freezing of Low Rank Adaptation in Parameter Efficient Fine-Tuning of Large Models." Proceedings of the 62nd Annual Meeting of the Association for Computational Linguistics (Volume 2: Short Papers). 2024.
> - LaMDA: Azizi, Seyedarmin, Souvik Kundu, and Massoud Pedram. "LaMDA: Large Model Fine-Tuning via Spectrally Decomposed Low-Dimensional Adaptation." Findings of the Association for Computational Linguistics: EMNLP 2024. 2024.
> - SSH (Shen et al., 2025b): Shen, Yixian, et al. "SSH: Sparse Spectrum Adaptation via Discrete Hartley Transformation." NAACL, 2025.
> - MaCP (Shen et al., 2025a): Shen, Yixian, et al. "MaCP: Minimal yet Mighty Adaptation via Hierarchical Cosine Projection." ACL, 2025.
>
> Notably, BiDoRA consistently outperforms all the baselines mentioned above, which showcases the effectiveness of our method.
>
> | Model     | SST-2    | MRPC     | CoLA     | QNLI     | RTE      | STS-B    | Avg.      |
> |-----------|----------|----------|----------|----------|----------|----------|-----------|
> | FF        | 94.8     | 90.2     | 63.6     | 92.8     | 78.7     | 91.2     | 85.22     |
> | BitFit    | 93.7     | **92.7** | 62.0     | 91.8     | **81.5** | 90.8     | 85.42     |
> | Adpt$^\text{D}$    | 94.7     | 88.4     | 62.6     | 93.0     | 75.9     | 90.3     | 84.15     |
> | LoRA      | 95.1     | 89.7     | 63.4     | 93.3     | 78.4     | **91.5** | 85.23     |
> | AdaLoRA   | 94.5     | 88.7     | 62.0     | 93.1     | 81.0     | 90.5     | 84.97     |
> | AFLoRA    | 94.1     | 89.3     | 63.5     | 91.3     | 77.2     | 90.6     | 84.33     |
> | LaMDA     | 94.6     | 89.7     | 64.9     | 91.7     | 78.2     | 90.4     | 84.92     |
> | VeRA      | 94.6     | 89.5     | 65.6     | 91.8     | 78.7     | 90.7     | 85.15     |
> | FourierFT | 94.2     | 90.0     | 63.8     | 92.2     | 79.1     | 90.8     | 85.02     |
> | SSH       | 94.1     | 91.2     | 63.6     | 92.4     | 80.5     | 90.9     | 85.46     |
> | MaCP      | 94.2     | 89.7     | 64.6     | 92.4     | 80.7     | 90.9     | 85.42     |
> | DoRA      | 94.9     | 89.9     | 63.7     | 93.3     | 78.9     | **91.5** | 85.37     |
> | BiDoRA    | **95.7** | 90.2     | **65.8** | **93.4** | 79.4     | 90.5     | **85.83** |

---

> > ### Author Response · Authors · 2025-06-27
> > **Official Rebuttal**
> >
> > > The background of this task is too long, occupying 60% of its content. Please make it more concise.
> >
> > We agree. We have moved the background descriptions of the datasets from the main experimental setup section to Appendix A to improve the flow of the main paper.
> > We also moved the background for the experiments in Section 5.5 to a new subsection in Appendix A.4 for the same reason.
> >
> > > There are still multiple issues on the mathematical notations and equations.
> >
> > We apologize for these errors.
> > We have revised the manuscript to fix the notation and equation issues. Specifically, we made the following changes based on your suggestions:
> >
> > 1.  We have rewritten all tensors, matrices, and vectors in bold to distinguish them from scalars.
> > 2.  For "$\|\cdot\|_c$ represents the vector-wise norm of a matrix", we have clarified in the main paper that we are using the $L_2$ norm.
> > 3.  For the loss function in Sec. 4.3, we changed the notation from $C$ to $\mathcal{L}$ for better consistency.
> > 4.  We added an equation number for the first equation in the optimization algorithm paragraph.
> >
> > > In the related work section, the adapter based PEFT methods are too old. The authors are suggested to discuss more recent PEFT methods in 2023-2024, and give their due credit to the adapter field.
> >
> > Thank you for this suggestion. As requested, we have added three recent adapter-based PEFT methods to the Related Work section in our updated version.
> >
> > > Caption of Fig.3. Please define $\Delta D$ and $\Delta M$ in the caption.
> >
> > We have added an explanation for these two terms in the caption as suggested.
> >
> > > Can some failure cases and limitation be discussed at the end of this paper?
> >
> > In the final "Conclusion and Future Work" section, we have added a paragraph discussing limitations related to training efficiency and the lack of a formal theoretical analysis, and we propose potential future work to address them.
> >
> > > In the Section E of the appendix, please explicitly mention which specific type of the hardware the authors use for the experiments.
> >
> > For a fair comparison, we used a single NVIDIA A100 GPU for all methods compared in this section. We have added this detail to Appendix E.
> >
> > ### Minor comments for improvement:
> >
> > We have fixed all your suggested changes.

---

> > > ### Comment · Reviewer_nEhv · 2025-06-28
> > > **Re: Official Rebuttal**
> > >
> > > Thanks for the rebuttal. My concerns have been well addressed, and I think this paper is in a good shape for acceptance.

---

> > > > ### Author Response · Authors · 2025-06-29
> > > > **Re: Official Rebuttal**
> > > >
> > > > We sincerely appreciate your thoughtful review and are grateful for your acknowledgment of our rebuttal. Thank you for your feedback and support for our paper.

---

### Review · Reviewer_t6jb · 2025-06-07

**Summary Of Contributions:**

This paper analyzes the coupled relationship between the direction and magnitude components in LoRA. Based on this insight, the authors propose BiDoRA, a bi-level optimization framework that decouples the optimization of magnitude and direction to improve performance and mitigate overfitting. The experimental results show that BiDoRA consistently outperforms DoRA and LoRA, similar to full fine-tuning.

**Audience:**

Yes

**Broader Impact Concerns:**

No broader impact concerns.

**Claims And Evidence:**

Yes

**Requested Changes:**

Please refer to the weaknesses. Specifically:

1. Improve the clarity and presentation of Figure 2.
2. Justify the use of the validation dataset for magnitude optimization and the training and validation datasets for retraining direction.
3. Specify explicit convergence criteria for both the search and retraining phases.
4. Include results of BiDoRA across different LoRA ranks to demonstrate robustness.
Clarify whether retraining the magnitude was considered, and discuss whether retraining only the direction leads to sub-optimal coupling. Address whether this choice impacts the validity of the search phase convergence.

**Strengths And Weaknesses:**

Strengths:
1. This paper provides a principled analysis of the coupled relationship between the direction and magnitude in LoRA and develops a novel algorithm to address the problem.
2. The proposed BiDoRA shows effective results in experiments, providing empirical validation on a wide range of tasks and model scales.

Weaknesses:
1. Figure 2 is not professional: axis arrows are missing, and there is an additional distracting symbol in the top right corner.
2. For updating the magnitude $M$, why is the validation dataset used rather than the training dataset? Besides, for retraining the direction $V$, why are the training and validation datasets used? The rationale should be better justified.
3. In Algorithm 1, it is unclear how the convergence of BiDoRA is determined. What is the specific stopping criterion for the search phase and retraining phase?
4. The experiments fix the LoRA rank to 4, but do not report BiDoRA’s performance under varying ranks. It would be informative to understand how the method scales with rank.
5. While retraining the direction improves BiDoRA’s performance, it incurs additional computational cost. Moreover, since the direction and magnitude are coupled, it is unclear why retraining the magnitude is not performed. If one component is updated, why not adjust the other? Does this mean the convergence condition of the search phase is insufficient or suboptimal?

---

> ### Author Response · Authors · 2025-06-27
> **Official Rebuttal**
>
> We thank you for your constructive feedback. We provide our response to your review as follows.
>
> > Improve the clarity and presentation of Figure 2.
>
> Thank you for this suggestion. We have improved the figure as recommended.
>
> > Justify the use of the validation dataset for magnitude optimization and the training and validation datasets for retraining direction.
> > Clarify whether retraining the magnitude was considered, and discuss whether retraining only the direction leads to suboptimal coupling. Address whether this choice impacts the validity of the search phase convergence.
>
> We apologize for the potential confusion. Using different data splits for different optimization levels is a common practice in the bi-level optimization literature, most notably in the DARTS method [1], where the architecture and sub-networks are learned using different dataset splits. Optimizing all variables in a single loop can result in an over-expressive network, as the selection variables tend to activate all sub-networks to maximize expressiveness, which in turn incurs severe overfitting.
> In contrast, training the sub-networks with the selection module fixed on the training split while validating the selection module's effectiveness on an unseen validation split effectively mitigates this risk.
>
> Our method follows a similar rationale: we treat the magnitude component as the "architecture" and the direction component as the "sub-network" and train them on separate datasets.
> As stated in the experimental setup, `BiDoRA does not use any additional data compared to other baselines, as we create the validation set for upper-level optimization by splitting the original training set with an 8:2 ratio for all tasks.`
> The training and validation sets are two non-overlapping splits of the original training set, which is precisely what our method requires.
>
> The rationale for the retraining stage is also analogous to neural architecture search: after we find the optimal magnitude (architecture) in the search phase, we freeze it and train the direction component (sub-network) from scratch. Here, all available data are used to *maximize data utilization*, as a larger dataset generally improves performance.
>
> These rationales are presented in the fourth paragraph of the Introduction and at the end of Section 4.
>
> To further address your concern, we have conducted an additional ablation study on retraining the magnitude with the direction frozen, compared to retraining the direction with the magnitude frozen, as in our original design. The results show the superiority of retraining the direction.
>
> | Method                        | MNLI   | SST-2  | MRPC   | CoLA   | QNLI   | QQP    | RTE    | STS-B  | Avg    |
> |-------------------------------|--------|--------|--------|--------|--------|--------|--------|--------|--------|
> | BiDoRA (retraining magnitude) | $87.0$ | $94.3$ | $89.1$ | $60.7$ | $92.7$ | $91.0$ | $73.4$ | $89.9$ | $84.8$ |
> | BiDoRA                        | $87.1$ | $94.4$ | $89.4$ | $61.3$ | $92.7$ | $90.6$ | $76.1$ | $90.1$ | $85.2$ |
>
> We have added this ablative counterpart to Table 4.
>
> > Specify explicit convergence criteria for both the search and retraining phases.
>
> In practice, the convergence of the search phase is determined by the upper level's evaluation metric. For the retraining phase, we follow a stopping criterion similar to DoRA's by observing the performance on a held-out testing set. We have added this explanation to the end of Section 4.
>
> > Include results of BiDoRA across different LoRA ranks to demonstrate robustness.
>
> As requested, we have conducted extra experiments across different rank settings in Appendix H.2, where BiDoRA consistently outperforms DoRA.
> Notably, BiDoRA consistently outperforms DoRA with ranks of $8$ and $16$ in addition to the rank of $4$ used in the main paper.
>
> | Model           | SST-2 | MRPC | CoLA | QNLI | RTE  | STS-B | Avg.              |
> |-----------------|-------|------|------|------|------|-------|-------------------|
> | DoRA ($r=8$)    | 94.9  | 89.9 | 63.7 | 93.3 | 78.9 | 91.5  | 85.37             |
> | BiDoRA ($r=8$)  | 95.7  | 90.2 | 65.8 | 93.4 | 79.4 | 90.5  | 85.83             |
> | DoRA ($r=16$)   | 94.8  | 90.4 | 65.6 | 93.1 | 81.9 | 90.7  | 86.08             |
> | BiDoRA ($r=16$) | 95.0  | 90.8 | 66.7 | 93.3 | 82.6 | 90.9  | 86.55             |
>
> [1] Hanxiao Liu, Karen Simonyan, and Yiming Yang. Darts: Differentiable architecture search. In International Conference on Learning Representations, 2018.

---

> > ### Comment · Reviewer_t6jb · 2025-07-11
> >
> > Thanks for the effort in rebuttal. I'm still interested in the setting of bi-level optimization in this paper. In the rebuttal, the authors claimed:
> >
> > 'treat the magnitude component as the 'architecture' and the direction component as the 'sub-network'
> >
> > May I ask whether the components of architecture and sub-network can be switched? It means:
> >
> > 'treat the direction component as the 'architecture' and the magnitude component as the 'sub-network'

---

> > > ### Author Response · Authors · 2025-07-12
> > > **Re: Official Comment by Reviewer t6jb**
> > >
> > > Dear Reviewer,
> > >
> > > Thank you for your question and interest in our work! As we mentioned in Section 4.3 (in the paragraph titled "A bi-level optimization framework"), we chose to set the magnitude component as the upper-level variable for the following two primary reasons.
> > >
> > > **Common practice in literature**
> > >
> > > In the bi-level optimization literature, the upper-level problem usually has significantly fewer parameters than the lower-level one.
> > > Our formulation adheres to this principle.
> > > The magnitude component (our upper level) has a complexity of $\mathcal{O}(k)$, whereas the direction component (our lower level) has a much higher complexity of $\mathcal{O}(dr+kr)$ (Assume the weight matrix $\mathbf{W} \in \mathbb{R}^{d \times k}$ and $r$ is the lora rank).
> > > This design is therefore consistent with standard practice.
> > >
> > > **Soft selection via magnitude**
> > >
> > > Our BiDoRA method is analogous to the DARTS method used in Neural Architecture Search (NAS), where subnets are chosen by a selection variable. In our framework, the magnitude vector functions as a "soft" selection variable for the direction matrix. It scales each direction (akin to a subnet), thereby controlling its contribution.
> > >
> > > To clarify this point further, recall that in DoRA, the reparameterization involves a magnitude vector $\mathbf{m} \in \mathbb{R}^{k}$ and a direction matrix $\mathbf{V} \in \mathbb{R}^{d \times k}$.
> > > Overall, we have $\mathbf{W} = \mathbf{m}\frac{\mathbf{V}}{\|\|\mathbf{V}\|\|_c}$ where $\mathbf{m}$ is elementwisely multiplied with the columns of $\mathbf{V}$.
> > > If the elements of $\mathbf{m}$ were constrained to be binary (i.e., $\{0, 1\}$), then $\mathbf{m}$ would act as a hard selection variable, performing the exact same selection procedure as in NAS by turning directions on or off. By relaxing this constraint to allow the elements of $\mathbf{m}$ to be real numbers, we achieve a similar outcome through a "soft selection" mechanism that up-weights or down-weights certain directions.
> > >
> > > We hope our clarification can address your question.
> > >
> > > Best regards,
> > >
> > > The Authors

---

### Review · Reviewer_oMbr · 2025-06-14

**Summary Of Contributions:**

This paper proposes an extension to DoRA by separately and iteratively optimizing the magnitude and direction of parameters in the PEFT setup. This approach is framed as a form of bi-level optimization, a technique commonly used in neural architecture search. The authors identify a limitation in the original DoRA method, namely the difficulty of estimating magnitude and direction simultaneously, and address it with their proposed optimization strategy. The method is evaluated on a range of NLP tasks, with experimental results demonstrating its effectiveness, though improvements are occasionally marginal. The paper also includes ablation studies that support the contribution of bi-level optimization and other proposed techniques.

**Audience:**

Yes

**Broader Impact Concerns:**

There is no broader impact concern.

**Claims And Evidence:**

Yes

**Requested Changes:**

Major requests
- Please provide a clearer explanation of the overfitting issue and the limitation in learning capacity associated with the simultaneous estimation of magnitude and direction. Additionally, explain how the proposed bi-level optimization approach addresses these problems, both conceptually and empirically.
- Clarify the distinctions between the proposed method and related work. While some differences are implied, an explicit comparison would help readers better understand the novelty of the approach.

Minor comments for improving the paper
- Section 4.1: The connection between the methods mentioned in the last two sentences and the proposed approach is unclear. Please elaborate on how these methods relate to the main contribution.
- Section 4.2: Orthogonal Regularization is introduced without sufficient context. Although its empirical effectiveness is shown in Table 4, the rationale behind its use is not discussed. Please explain its role and importance in the proposed method.
- Section 4.3: The explanation of the "Optimization Algorithm" section, particularly regarding the analytical solution for $\mathcal{V}^{\ast}(\mathcal{M})$, is confusing. Since neural networks generally do not yield analytical solutions due to their non-linear nature, this claim requires clarification. I may be misunderstanding the point, so a more detailed explanation would be helpful.
- Section 5: There is no discussion of training cost. Since LoRA, DoRA, and other PEFT methods are typically valued for their efficiency and lightweight training requirements, it would be beneficial to include a comparison or analysis of training costs for the proposed method.

**Strengths And Weaknesses:**

Strengths
- PEFT is widely adopted across various large models, including LLMs, making this a highly relevant and important topic for the machine learning community.
- The use of bi-level optimization to separately estimate magnitude and direction parameters is a novel contribution.
- The experimental evaluation is thorough, and the ablation studies are well-designed and informative.

Weaknesses
- The motivation for separately estimating magnitude and direction parameters remains unclear. While the paper identifies two issues with simultaneous estimation—(1) increased expressiveness potentially leading to overfitting, and (2) a coupled update pattern that may limit learning capacity—it does not provide sufficient explanation or evidence to support these claims.
- Specifically, it is not demonstrated whether overfitting or limited learning capacity actually occur in practice, nor is it clearly shown how bi-level optimization addresses these issues, either logically or through experimental validation. This lack of clarity is a major concern.
- In Section 2, the relationship between the proposed method and related work is not clearly delineated. While some differences can be inferred, it would strengthen the paper to make these distinctions explicit.

---

> ### Author Response · Authors · 2025-06-27
> **Official Rebuttal**
>
> We thank you for your constructive feedback. We provide our response to your review as follows.
>
> > Please provide a clearer explanation of the overfitting issue and the limitation in learning capacity associated with the simultaneous estimation of magnitude and direction. Additionally, explain how the proposed bi-level optimization approach addresses these problems, both conceptually and empirically.
>
> Thank you for raising this point. Figure 2 in the main paper shows that BiDoRA achieves a smaller performance gap between the training and test sets. To provide a more direct empirical analysis, we have conducted an additional experiment on the quantitative performance gap, presented in Appendix H.3. Using the RoBERTa-base model, we calculate the gap as follows:
> For the training set metric, we use the moving average of the per-batch metric with a decay ratio of $0.99$. In BiDoRA, which has two training loops, we calculate the training metric for each and then compute a weighted average based on the data split ($inner:outer=8:2$). Specifically, the performance gap for DoRA is $training-testing$, while for BiDoRA it is $0.8\times inner+0.2\times outer- testing$.
>
> | Method | SST-2   | MRPC    | CoLA     | QNLI    | RTE      | STS-B   | Avg.    |
> |--------|---------|---------|----------|---------|----------|---------|---------|
> | DoRA   | 2.0     | 9.5     | 32.5     | 6.6     | 18.0     | 8.8     | 12.9    |
> | BiDoRA | **1.7** | **7.0** | **23.3** | **0.2** | **14.0** | **4.7** | **8.5** |
>
>
> The results show that the performance gap for BiDoRA is consistently lower than that of DoRA across all datasets, which empirically demonstrates the overfitting issue in the DoRA method and shows that our BiDoRA effectively addresses this issue.
>
> > Clarify the distinctions between the proposed method and related work. While some differences are implied, an explicit comparison would help readers better understand the novelty of the approach.
>
> Inspired by NAS, where a bi-level approach is utilized to learn architecture and sub-network weights using separate data splits to prevent overfitting, we adapt the BLO framework to the problem of parameter-efficient fine-tuning (PEFT), specifically for the weight-decomposed adaptation introduced by DoRA. Unlike in NAS, where BLO searches for a network architecture, BiDoRA repurposes it to decouple the optimization of a weight matrix's two components: its magnitude and direction.
>
> This approach marks a significant difference from previous PEFT methods like LoRA and DoRA, which optimize all their trainable parameters simultaneously on a single dataset. In this work, we extend the application scenarios of gradient-based BLO to develop a robust and effective parameter-efficient fine-tuning method for pre-trained models. By assigning the magnitude and direction components to different optimization levels with distinct data splits, BiDoRA creates a decoupled and more flexible updating pattern that better mitigates overfitting and more closely resembles the learning behavior of full fine-tuning.
>
> We have added this clarification at the end of Section 2.2.
>
> > Section 4.1: The connection between the methods mentioned in the last two sentences and the proposed approach is unclear. Please elaborate on how these methods relate to the main contribution.
>
> While this work focuses on the empirical validation of BiDoRA, our choice of this optimization strategy is grounded in established theoretical research, including [1], [2], and references therein. The convergence properties of similar gradient-based bi-level algorithms have been previously analyzed, providing confidence in the stability of our training procedure. Furthermore, the ability of such frameworks to improve generalization—a core objective of BiDoRA—has also been formally studied in [3], demonstrating a rationale that our approach can mitigate overfitting.
>
> We have added this clarification at the end of Section 4.1.
>
> [1] Fabian Pedregosa. Hyperparameter optimization with approximate gradient. In International conference on machine learning, pp. 737–746. PMLR, 2016.
>
> [2] Aravind Rajeswaran, Chelsea Finn, Sham M Kakade, and Sergey Levine. Meta-learning with implicit gradients. Advances in neural information processing systems, 32, 2019.
>
> [3] Fan Bao, Guoqiang Wu, Chongxuan Li, Jun Zhu, and Bo Zhang. Stability and generalization of bilevel programming in hyperparameter optimization. Advances in neural information processing systems, 34:4529–4541, 2021.

---

> > ### Author Response · Authors · 2025-06-27
> > **Official Rebuttal**
> >
> > > Section 4.2: Orthogonal Regularization is introduced without sufficient context. Although its empirical effectiveness is shown in Table 4, the rationale behind its use is not discussed. Please explain its role and importance in the proposed method.
> >
> > A central goal of BiDoRA is to effectively learn the two disentangled components of a weight update: magnitude and direction. The direction component, $\Delta \mathbf{V}$, is tasked with finding a low-rank basis for the update directions. To maximize the expressive power of this component and prevent overfitting, the basis vectors (i.e., the columns of the direction matrix) should be *as diverse and non-redundant as possible*. Orthogonality is an ideal property to enforce this diversity. Enforcing orthogonality on the direction vectors constrains them to represent distinct, independent pathways for updates, ensuring that the limited parameter budget of the low-rank matrix is used efficiently to explore the solution space. This constraint helps counteract the tendency of models to learn correlated, redundant features, which can contribute to overfitting.
> >
> > We have added this clarification in Section 4.2.
> >
> > > Section 4.3: The explanation of the "Optimization Algorithm" section, particularly regarding the analytical solution for $\mathcal{V}^*(\mathcal{M})$, is confusing. Since neural networks generally do not yield analytical solutions due to their non-linear nature, this claim requires clarification. I may be misunderstanding the point, so a more detailed explanation would be helpful.
> >
> > Solving the bi-level optimization problem stated in Section 4.3 presents a significant challenge: computing the gradient of the upper-level loss $\mathcal{L}_{val}$ with respect to the magnitude component. This is because the true gradient for the upper level depends on the optimal solution of the lower-level problem. For deep neural networks, the lower-level objective is a non-convex function, and finding its true optimal solution would require running the optimization process to full convergence. Performing this complete inner optimization for every single update of the upper-level variable is computationally intractable. To make this process feasible, we adopt a common and effective strategy from the BLO literature and approximate the solution of the lower level with a one-step gradient descent update.
> >
> > We have added this clarification in Section 4.3.
> >
> > > Section 5: There is no discussion of training cost. Since LoRA, DoRA, and other PEFT methods are typically valued for their efficiency and lightweight training requirements, it would be beneficial to include a comparison or analysis of training costs for the proposed method.
> >
> > An analysis of the training cost is available in Appendix E. The results reveal that the total training time for BiDoRA is approximately 1.64 times that of DoRA, a training cost that remains comparable to the baselines.
> >
> > We would like to emphasize that while BiDoRA has a slightly increased training cost compared to DoRA, its significant performance improvements and novel approach to addressing overfitting represent a worthwhile trade-off that is generally acceptable in practice.

---

### Author Response · Authors · 2025-06-27
**General Response**

Dear reviewers,

To address your concerns, we have revised our paper and uploaded the updated version, with major changes highlighted in red. The key changes are as follows:

1.  In Appendix H.1, we have added extra experiments with a wide range of more recent comparison methods. BiDoRA consistently outperforms these baselines, showcasing its superiority.
2.  In Appendix H.2, we have also conducted extra experiments across different rank settings, where BiDoRA consistently outperforms DoRA.
3.  In Appendix H.3, we quantitatively measure the performance gap between the training and test sets. The results show that the gap for BiDoRA is consistently lower than that of DoRA across the GLUE benchmark, providing evidence that the overfitting issue occurs in DoRA and BiDoRA addresses this.
4.  In Appendix I, we have added a visualization of the direction matrix eigenvalues (Figure 5) to demonstrate the effectiveness of orthogonal regularization more clearly.
5.  In Appendix E, we re-evaluate the training cost of BiDoRA compared to baselines in terms of total training time, instead of only the per-step cost presented previously. The total training time for BiDoRA is approximately $1.64$ times that of DoRA, a training cost that remains comparable to the baselines.

Thank you again for your helpful reviews. We welcome any further questions or concerns you may have.

Best regards,

Authors

---

### Decision · Action_Editor_G4s3 · 2025-07-31

**Recommendation:** Accept as is

**Additional Comments:**

I encourage the authors to adapt the manuscript based on the feedback of reviewers, but I don’t see issues that warrant delaying acceptance.

**Audience:**

Yes

**Audience Explanation:**

The manuscript presents an insightful study of the original weight-decomposed low-rank adaptation (DoRA) mechanism. It reveals that simultaneously optimizing magnitude and direction of parameters causes a coupled gradient update pattern which limits capacity.
Motivated by neural architecture search, the authors propose a bi-level optimization approach, bi-Level optimization-based weight-decomposed low-rank adaptation (BiDoRA), to address this challenge. The manuscript also shows how across several benchmarks BiDoRA more closely matches full-finetuning compared to DoRA and multiple other baseline parameter efficient finetuning (PEFT) methods.

PEFT methods are of general interest to the community and DoRA has received considerable attention, suggesting a well-motivated, improved version would be of interest to at least some audience members.

**Claims And Evidence:**

Yes

**Claims Explanation:**

Reviewers generally agreed that the provided evidence is convincing and backs up the claims made in the manuscript. Reviewer nEhv had initial concerns with the lack of visual analysis and empirical evidence to demonstrate the effectiveness of orthogonal regularization, but the rebuttal adequately addressed these concerns by providing additional visual evidence and referring to the rationale for existing visualizations.